# How Many Code and Test Cases Are Enough? Evaluating Test Cases Generation from a Binary-Matrix Perspective

**Xianzhen Luo[1†]  Jinyang Huang[2†]  Wenzhen Zheng[3]  Qingfu Zhu[1*]  Mingzheng Xu[1]  Yiheng Xu[4]  Yuantao Fan[3]  Wanxiang Che[1*]**
[1] Harbin Institute of Technology  [2] Central South University  [3] StepFun  [4] Peking University
{xzluo,qfzhu,car}@ir.hit.edu.cn  {hjy.tsuki}@csu.edu.cn

## Abstract

Evaluating test cases automatically generated by Large Language Models (LLMs) is a critical yet challenging task. Existing benchmarks often evaluate the exclusion ratio on large, unstructured collections of wrong codes, suffering from high computational costs and score inflation. Furthermore, they inadvertently reward generators that detect common, trivial bugs, while failing to penalize their inability to identify rare yet critical faults. In this work, we connect two fundamental questions: (1) What is the minimal set of wrong codes sufficient to represent the entire error space? and (2) What is the minimal set of test cases needed to distinguish them? We introduce a novel framework that formalizes benchmark construction as finding an optimal diagnostic basis in a binary code-test matrix, where rows represent wrong codes and columns represent test case results. The rank of this matrix specifies the minimal number of independent error patterns (wrong codes) and provides a tight upper bound on the number of test cases required for complete fault coverage. Our objective is to identify a basis of size equal to the matrix rank that maximizes internal diversity. To tackle this NP-hard problem, we propose WrongSelect, an efficient approximation algorithm to select maximally diverse wrong codes. Applying this framework to millions of competitive programming submissions, we construct TC-Bench, a compact, diverse, and inflation-resistant benchmark. Extensive experiments show that even the most advanced test case generation methods achieve only 60% exclusion rates on TC-Bench, exposing a significant gap in their diagnostic power and highlighting substantial room for future improvement. Our dataset is available at: https://huggingface.co/datasets/Luoberta/TC-Bench and our code is at: https://github.com/Luowaterbi/TC-Bench.

## 1 Introduction

The capability of Large Language Models (LLMs) in solving algorithmic coding problems is a key measurement of their intelligence (OpenAI et al., 2024; 2025; Jain et al., 2024). The evaluation of code solutions relies heavily on test cases. Golden Test cases (GTs), created by problem authors and continually refined and expanded by experts, are considered a boundary-condition set equivalent to the correct solution. A solution is deemed correct only if it passes GTs. Current Code Reinforcement Learning with Verifiable Rewards (RLVR) methods similarly rely on test cases to compute rewards, placing substantial demands on the comprehensiveness of test cases (Le et al., 2022; Guo et al., 2025; Team et al., 2025; Zeng et al., 2025a). As shown in Figure 1 (a), the GT of a graph theory problem should encompass various graph sizes and structures, such as chain, tree, and star. Failure to cover all scenarios will compromise the reliability and lead to the false positive problem.

GTs consist of a few simple public test cases intended to clarify the problem and a larger set of private test cases used to assess correctness. However, these critical private test cases are scarce

---

*Corresponding author.
†Equal contribution.

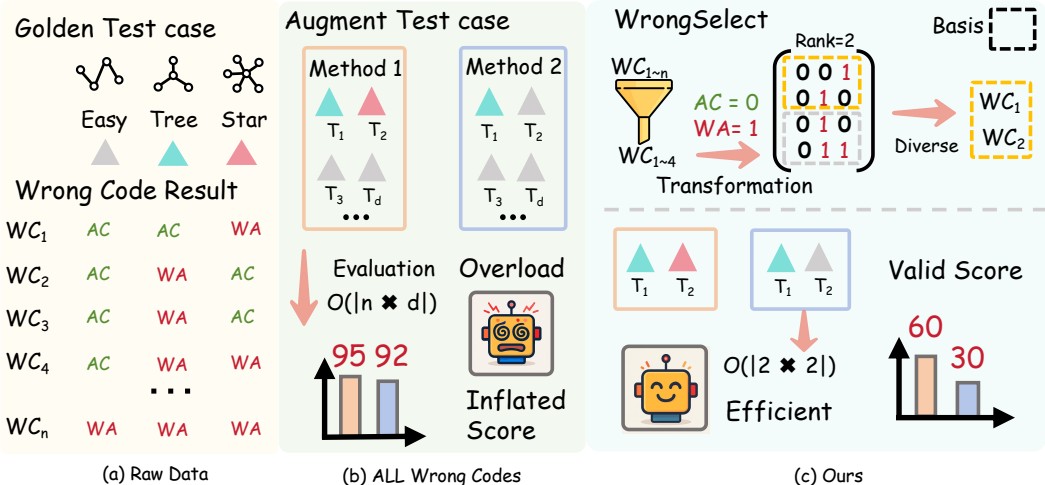

Figure 1: A comparison of two evaluation frameworks for Augment Test cases (ATs). Both frameworks start from the same raw data (a), which consists of many wrong codes (WCs) and their execution results on Golden Test cases (GTs). (b) The naive evaluation utilizes the full set of WCs and an unprincipled number of ATs, suffers from prohibitive computational costs, and leads to inflated scores. (c) In contrast, our proposed framework first processes this data with WrongSelect to select a compact yet representative diagnostic basis (TC-Bench). Evaluation using this basis is not only highly efficient but also yields more valid scores.

and expensive to create. To address this challenge, existing methods either manually construct test cases (Khan et al., 2023) or automatically augment test cases (ATs) using LLMs (Cao et al., 2025; Ma et al., 2025b; Yang et al., 2025; Wang et al., 2025c). The emergent methods introduce the need to evaluate their quality. The evaluation includes ensuring that their ATs are valid (passing correct codes) and useful (excluding wrong codes (WCs) ). Since many methods are seeded with correct codes, their ATs are naturally valid. Thus, the core challenge shifts to assessing their usefulness. The straightforward approach is to collect as many wrong codes as possible and evaluate all ATs to determine how many WCs they can exclude. However, this incurs immense computational costs and suffers from inflated scores as shown in Figure 1 (b). This cost, a product of the number of ATs and WCs, can be prohibitively high. Furthermore, one WC doesn't equal one kind of error. Indeed, the population of WCs is dominated by numerous trivial or repetitive errors, with only a few representing core, hard-to-detect faults (Figure 1 (a) ). A mediocre method that only identifies common errors can thus achieve a score similar to a superior method that finds rare corner cases, as the small number of critical faults gets statistically overwhelmed. Consequently, this diminishes the benchmark's discriminative power. Conversely, some heuristic methods selecting a small subset of hard-to-filter errors yield overly sparse evaluations (Cao et al., 2025), unable to continuously reflect model capabilities.

These limitations raise fundamental questions: *What constitutes an efficient and informative collection of WCs for evaluating ATs? What principles should govern its size and member selection?* The dual relationship between test cases and code also leads to another critical question: *How many test cases are necessary to comprehensively define the solution space for a given problem?*

We propose that **an ideal WCs set should** neither be heuristically nor randomly selected, but should **be a compact and diverse set of WCs that acts as a diagnostic basis, effectively spanning all unique error patterns of the problem.** We propose to interpret the execution outcomes of WCs across GTs as a mapping from abstract reasoning errors to observable behavioral patterns. In this binary representation, the accepted (AC) is denoted as 0 and wrong answer (WA) as 1. Each WC is thus represented as a binary vector, and the entire collection forms a Code-Test binary matrix. The matrix rank quantifies the maximum number of distinct error patterns present among WCs. Moreover, it provides an upper bound on the minimal number of test cases required to distinguish these error patterns. However, a matrix can produce multiple possible bases. An optimal diagnostic basis

should consist of WCs representing minimally overlapping error patterns to maximize diagnostic breadth and information efficiency. Bases containing many similar WCs with highly overlapping error patterns suffer from redundancy, thus reducing discriminative power. As finding the most diverse basis is NP-hard, we design WrongSelect, a greedy-based efficient approximation algorithm that iteratively selects WCs that maximize diversity at each step, yielding the final basis.

To construct our high-quality benchmark, we collect numerous problems with their GTs and user submissions from prestigious algorithm competitions like USACO, NOI, and ICPC. We rigorously filter submissions, retaining only those with complete execution results on GTs. Next, we transform the codes for each problem into a binary matrix and calculate its rank to characterize the error pattern complexity. Then, we employ WrongSelect to efficiently select a maximally diverse set of WCs, constructing a structured diagnostic basis (Figure 1 (c) ). Last, we meticulously review, standardize, and translate all problem descriptions into English to ensure consistency and quality. The resulting benchmark, named TC-Bench, contains 877 problems with a total of 9347 WCs. The final set of WCs constitutes less than 2% of the original submissions. This reduction, combined with the principled number of the necessary test cases, can lead to a near-quadratic decrease in evaluation cost, dramatically improving efficiency. To validate TC-Bench, we reproduce and evaluate 5 common test-case generation methods (Jain et al., 2024; Zeng et al., 2025b; Zhang et al., 2023; He et al., 2025; Gu et al., 2024) on 13 leading LLMs (DeepSeek-AI et al., 2024; Int; Hui et al., 2024). Experimental results show that even the state-of-the-art method Claude4-Thinking with LCB achieve only approximately 60% performance. By eliminating redundant error patterns and surfacing critical corner cases, TC-Bench ensures that a method's ability to handle these challenges is directly reflected in its score. This directly prevents the score inflation that plagues less-curated benchmarks.

Our contributions can be summarized as follows:

- We propose a novel framework based on matrix rank that, for the first time, unifies two fundamental questions: the minimal number of wrong codes needed for evaluation and the minimal number of test cases needed for coverage. This framework provides a principled method for constructing a structured diagnostic basis.

- We construct and release TC-Bench, a compact and diverse benchmark built on our theory. By design, TC-Bench has a high signal-to-noise ratio, enabling efficient, reliable, and inflation-resistant evaluation of test case generation methods.

- Through extensive empirical experiments, we uncover significant deficiencies in current mainstream test-case generation methods and LLMs when dealing with complex error patterns, providing clear guidance for future research.

## 2 METHODOLOGY

This section details our principled approach to constructing TC-Bench. We first formalize the problem as finding a maximally diverse basis within a binary Code-Test matrix (Section 2.1). Recognizing this problem as NP-hard, we then propose WrongSelect, a greedy approximation algorithm for this task (Section 2.2). Finally, we detail the data processing pipeline used to apply this framework in practice to build TC-Bench (Section 2.3).

### 2.1 PROBLEM FORMULATION

Identifying diverse underlying errors in a vast collection of WCs would require immense manual effort from algorithm experts, which is clearly infeasible. Therefore, the key challenge is to finding a formal transformation that can equivalently represent the diversity of underlying errors.

Our inspiration comes from how codes are evaluated. A code is considered correct if and only if it passes GTs, which are assumed to cover all problem requirements and boundary conditions, thereby defining the solution space. For any code, we can get its result on GTs. For example, the result ["AC", "WA", "WA"] represents a code that passes the first case but fails the other two. Such a result sequence can be regarded as a behavioral mapping or a failure signature, translating the abstract erroneous reasoning of a code into a concrete pattern within the solution space. Collecting all such signatures across codes allows us to construct an empirical space of failure modes for a problem.

However, this raw space is highly redundant: it contains identical signatures, and some patterns may simply be combinations of other ones. To extract a compact and informative benchmark from this landscape, a structured analytical tool is required.

**Binary Matrix Representation**  We formalize this space of failures as a binary matrix $M$ of size $n \times d$, where $n$ is the number of WCs and $d$ is the number of GTs. Each entry $M_{ij}$ is defined as:

$$M_{ij} = \begin{cases} 1 & \text{if the } i\text{-th WC fails on the } j\text{-th test case,} \\ 0 & \text{if the } i\text{-th WC passes the } j\text{-th test case.} \end{cases}$$

Each row vector $\mathbf{r}_i$ of $M$ thus represents the failure signature of the $i$-th WC. For instance, signature ["AC", "WA", "WA"] becomes the binary vector $[0, 1, 1]$.

**Optimization Objective**  With this binary Code-Test matrix in place, our task reduces to a selection problem: how to choose from the $m$ failure signatures a representative and compact subset $\mathcal{I}$ to serve as our benchmark. An ideal subset $\mathcal{I}$ must satisfy the following two requirements. **Completeness and Irredundancy**. The selected set $\mathcal{I}$ should capture the full complexity of $M$ without redundancy. In linear algebra, this corresponds precisely to a basis. Concretely, $\mathcal{I}$ must be a row basis, i.e., the row vectors in $\mathcal{I}$ are linearly independent and their number $|\mathcal{I}|$ equals the rank of $M$. This constraint guarantees that the number of selected WCs is neither too many nor too few, but exactly sufficient to span all distinct error modes. Notably, since the row rank equals the column rank, this same value $|\mathcal{I}|$ also provides another important insight: it constitutes a theoretical upper bound on the minimum number of test cases required to distinguish all independent error modes. **Diversity**. Multiple bases may satisfy the rank condition. Ideally, a perfect basis would consist of mutually orthogonal failure signatures, meaning each error mode is completely independent and contributes a unique dimension. However, in real-world error data, this kind of orthogonal basis rarely exists. Our practical goal is therefore to find a basis that approximates orthogonality by maximizing the diversity among its members (i.e., minimizing their overlap). To measure the overlap between two signatures, we adopt the Jaccard similarity, which quantifies the ratio of jointly failed test cases to the total failed cases across both signatures. A lower Jaccard score indicates lower similarity. Formally:

$$J(\mathbf{r}_i, \mathbf{r}_j) = \frac{\mathbf{r}_i \cdot \mathbf{r}_j}{\|\mathbf{r}_i\|_1 + \|\mathbf{r}_j\|_1 - \mathbf{r}_i \cdot \mathbf{r}_j}$$

where $\mathbf{r}_i \cdot \mathbf{r}_j$ counts the jointly failed test cases (intersection) and $\|\mathbf{r}\|_1$ is the total number of failed tests for a signature (size of the set).

Beyond pairwise similarity, we must assess the diversity of the entire basis $\mathcal{I}$. We therefore define our global objective as minimizing the average pairwise Jaccard similarity among all members of $\mathcal{I}$:

$$\min_{\mathcal{I}} F(\mathcal{I}) = \frac{1}{\binom{|\mathcal{I}|}{2}} \sum_{\mathbf{r}_i, \mathbf{r}_j \in \mathcal{I}, i < j} J(\mathbf{r}_i, \mathbf{r}_j)$$

In summary, our problem is formalized as follows: given a binary matrix $M$, find a row basis $\mathcal{I}$ that minimizes the average pairwise Jaccard similarity $F(\mathcal{I})$. This is a combinatorial optimization problem known to be NP-hard. In the next section, we present a greedy algorithm, WrongSelect, designed to efficiently approximate this solution.

## 2.2 WrongSelect

### 2.2.1 Principled Pre-filtering

The quality of the final basis critically depends on the quality of the candidate pool. In practice, raw data often contains noise, such as problems lacking sufficient WCs or WCs failing on all test cases. To address this, pre-filtering is designed to systematically remove these noise at both the problem level and the code level.

**Problem-Level Filtering via Column Analysis**  In practice, we observe that some $M$ contain columns filled entirely with "1" as shown in Figure 2. This indicates that all WCs fail in one case. The analysis on a subset shows that this phenomenon arises from three main causes: (1)

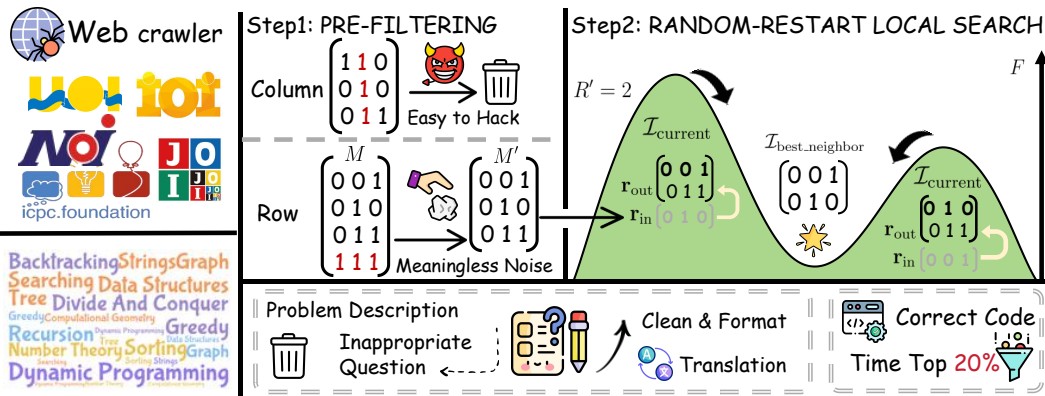

Figure 2: An overview of the TC-Bench construction pipeline. It begins with raw data collection, followed by a two-step WrongSelect working on the transformed binary matrix $M$. Step 1 pre-filters the problems with an all- "1" column and removes codes whose rows have too many "1"s. Step 2 samples numerous initial bases $\mathcal{I}_{current}$ from the filtered $M'$ and iteratively minimizes the diversity score by swapping internal and external rows. The best local optimum is chosen to approximate the global optimum. Concurrently, problem descriptions are standardized and correct codes are sampled from the top 20% performers, ensuring the overall quality of TC-Bench.

GT exhibits incremental difficulty (e.g., gradually stricter constraints on time or space complexity); (2) the number of WCs for the problem is insufficient; or (3) the problem or GT is overly simple, involving only a single extreme scenario. Although the first case is reasonable, it is relatively rare, and manually distinguishing it is prohibitively costly. More importantly, all-ones columns open the door to hack scores. Therefore, to ensure the diagnostic value of each problem, we exclude all problems containing all-ones columns from our dataset. This excludes about 5% of raw problems.

**Code-Level Filtering via Row Analysis** Another observation is that some WCs fail on an excessively high proportion of GTs. Such WCs typically pass only the public test cases while failing almost all private ones. They act as strong background noise: any mediocre test set can easily eliminate them, leading to inflated evaluation scores and severely diminishing the discriminative power of the benchmark. To mitigate this, we compute the failure rate of each row, which is defined as the proportion of 1's relative to $d$. Accordingly, we set the filtering threshold $\tau = 80\%$. A WC exceeding $\tau$ is highly likely to fail all private test cases. Such WCs generally correspond to trivial or common error patterns, and removing them helps benchmark to diagnose more nuanced and complex failure modes. This removes 13% of raw WCs and $M$ turns into $M'$.

As the final quality control step, we exclude all $M'$ with rank less than 5 ($R' < 5$). A low rank indicates insufficient diversity in error patterns and is not suitable to be used in a benchmark. Only matrices $M'$ that pass all these filtering stages are considered qualified candidates and proceed to the subsequent basis selection process.

### 2.2.2 RANDOM-RESTART LOCAL SEARCH

On the filtered matrix $M'$, our objective is to select a basis $\mathcal{I}$ that achieves the lowest possible $F(\mathcal{I})$. We adopt a local search optimization strategy to approximate the optimal basis.

Starting from a complete but randomly chosen initial basis, the algorithm iteratively improves the basis by performing local modifications. Specifically, it explores the neighborhood of the current basis, defined as all new bases that can be obtained by a single swap operation (exchanging one member inside the basis with one outside). If there exists a neighbor that achieves better diversity (lower $F(\mathcal{I})$), the basis is replaced by the best neighbor, and the process repeats. This iterative improvement continues until no better neighbor exists, i.e., the current basis converges to a local optimum. To mitigate the risk of being trapped in poor local optima due to initialization, we employ

a random-restart mechanism. The local search process is repeated multiple times from different random starting points, and finally, the best solution among all local optima is selected as the output.

Take "Step2" in Figure 2 as an Example. $M'$ has a rank of $R' = 2$. Assume the initial random basis is $\mathcal{I}_{\text{current}} = [[0, 0, 1], [0, 1, 1]]]$, with a diversity score of $F(\mathcal{I}_{\text{current}}) = 0.5$. The only external candidate is the vector $M' - \mathcal{I}\text{current} = [0, 1, 0]$. The algorithm then explores the neighborhood of $\mathcal{I}_{\text{current}}$. It first considers swapping the internal vector $\mathbf{r}_{\text{out}} = [0, 0, 1]$ with the external vector $\mathbf{r}_{\text{in}} = [0, 1, 0]$. The resulting set, $[[0, 1, 0], [0, 1, 1]]$, is a valid basis, but its score $F = 0.5$ provides no improvement. Next, consider swapping $\mathbf{r}_{\text{out}} = [0, 1, 1]$ with $\mathbf{r}_{\text{in}} = [0, 1, 0]$. This produces a better basis $\mathcal{I}_{\text{temp}} = [[0, 0, 1], [0, 1, 0]]$. Its diversity score is $F(\mathcal{I}_{\text{temp}}) = 0$. After evaluating all neighbors, since a better neighbor is found, the algorithm updates its state: $\mathcal{I}_{\text{current}} \leftarrow [[0, 0, 1], [0, 1, 0]]$. A new search iteration begins from this basis. As this basis is now perfectly diverse ($F = 0$), no further swaps can improve the score, so the algorithm has converged to a local optimum. This result is saved, and the random-restart mechanism initiates a new search from another random starting point. Algorithm 1 in Appendix A.2 illustrates the pseudo code with a detailed explanation.

Although the nested structure suggests high theoretical complexity, in practice the algorithm converges rapidly in both the inner and outer loops. Moreover, several parts of the procedure can be parallelized easily, making the overall runtime highly efficient.

## 2.3 BENCHMARK CONSTRUCTION

Evaluating test cases requires not only wrong code, but also first generating them from problem descriptions and validating them against correct code. This section details the full pipeline of data collection, filtering, and cleaning used to construct our benchmark.

**Raw Data** The raw data comes from top-tier programming contests and high-quality training sets, including USACO, IOI, and ICPC. In total, it initially contains 3,321 problems and 2,230,009 submissions. We retain only problems for which the full execution results of WCs on GTs are available. After this step, we obtain 1,763 problems, containing 15,457 correct codes and 554,056 WCs.

**Problem Description** To ensure fair and consistent problem comprehensions, we apply rigorous standardization to problem descriptions. We first remove problems that heavily rely on images, cannot be automatically evaluated (e.g., interactive problems, multi-solution tasks), or require highly constrained runtime environments. We then clean the statements by removing source tags, URLs, and HTML, as well as rewriting non-standard mathematical formulas. Finally, we employ GPT-4o to translate non-English problems and manually proofread to ensure semantic consistency.

**Wrong Code** To ensure consistency of the evaluation environment and avoid noise introduced by environment-specific factors, we retain only C++ submissions labeled as WA, including 1,698 problems and 282,458 WCs. Next, our principled pre-filtering leaves 1,133 problems with 33,846 WCs. For each problem, we perform random-restart local search with both outer and inner loops set to 1000 iterations. Figure 9 shows that loops converge rapidly, demonstrating the efficiency of our method. Ultimately, 13,400 wrong codes constitute the maximally diverse basis for all problems. Figure 10 illustrates the distribution of WCs per problem before and after WrongSelect.

**Correct Code** Since correct codes are consistent with GT, their primary differences lie in runtime and memory consumption. In Section 4, we show that overly loose or overly strict sets of correct codes can bias evaluation results. Therefore, for each problem, we randomly select 8 correct submissions from the top 20% after min–max normalization of runtime.

Through this principled pipeline, we ultimately construct TC-Bench, a high-quality diagnostic benchmark with 877 standardized problems, 9347 core WCs, and 7016 correct submissions. More details regarding the construction process are available in Appendix B.1. Furthermore, we present a case study in Appendix C to empirically validate the practical effectiveness of WrongSelect.

## 3 EXPERIMENT

After constructing TC-Bench, this section presents the experimental design and evaluation results of different test case generation methods.

### 3.1 EVALUATION SETUP

#### 3.1.1 MODELS & METHODS

**Models** We evaluate SOTA LLMs via API, including GPT-4o, Claude-Sonnet-4, Claude-Sonnet-4-Thinking, DeepSeek-V3, Qwen-Coder-Plus, and Qwen3-235B-A22B. We also evaluate Qwen-2.5 and Qwen-2.5-Coder families of varying sizes (7B, 14B, 32B). Due to space constraints, the results for the 7B and 14B LLMs are presented in Appendix A.3. We note that DeepSeek-R1 struggles to reliably generate test cases. Therefore, its results are excluded from the main experiments but discussed in Appendix A.5. In total, we evaluate 13 LLMs.

**Methods** Based on whether correct code is available during generation, methods can be categorized into two classes. The first class does not rely on correct code. **CRUX** (Gu et al., 2024) directly generates inputs and outputs. **PSEUDO** (Jiao et al., 2024) generates both inputs and candidate solutions, then obtains outputs by executing the solutions and taking the majority-voted output as the result. Going further, **ALGO** (Zhang et al., 2023) prompts the LLM to produce input generators (execute to obtain inputs) and a brute-force oracle solution (lower the difficulty).

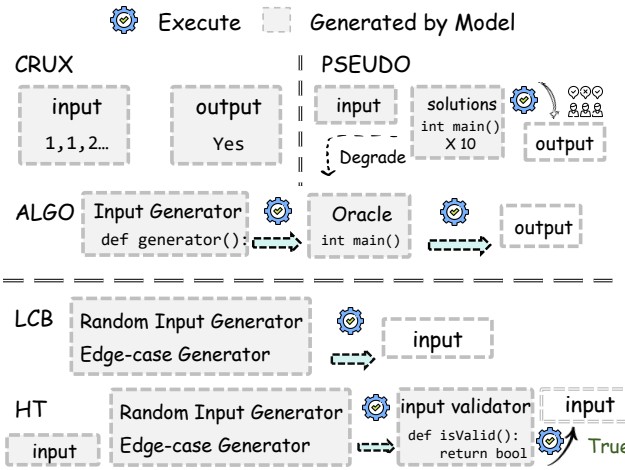

When the correct code is available, output correctness can be guaranteed by executing the inputs on it. Live-CodeBench (**LCB**) (Jain et al., 2024) requires LLM to generate both multiple random and edge-case input generators. It should be noted that we select one representative implementation for each category, and the other variants are in Appendix A.1.

Figure 3: CRUX, PRESUDO, and ALGO construct the output, while LCB and HT depend on the correct code to generate the output.

#### 3.1.2 PIPELINE & METRICS

**Test Case Generation** For each problem in TC-Bench, ATs are first generated by the evaluated methods. For methods that do not rely on correct code, only cases accepted by all correct code are considered valid. We define PassRate as the proportion of valid cases among all generated cases. Formally, for a set of problems $\mathcal{Q}$: $PassRate = \frac{1}{|\mathcal{Q}|} \sum_{q_i \in \mathcal{Q}} \left( \frac{1}{|\mathcal{T}_{q_i}|} \sum_{t_j \in \mathcal{T}_{q_i}} \text{IsValid}(t_j) \right)$, where $\mathcal{T}_{q_i}$ is ATs for problem $q_i$, and $\text{IsValid}(t_j)$ is 1 if test $t_j$ is valid, and 0 otherwise.

**Wrong Code Execution** To measure the effectiveness of the valid ATs, we define HackRate. A WC from TC-Bench is considered excluded if it fails on at least one valid AT. All failure types (e.g., WA, Time Limit Exceeded (TLE), RE (Runtime Error)) are counted as successful exclusion. The HackRate represents the proportion of WCs that are successfully excluded. Formally: $HackRate = \frac{1}{|\mathcal{Q}|} \sum_{q_i \in \mathcal{Q}} \left( \frac{1}{|\mathcal{W}_{q_i}|} \sum_{w \in \mathcal{W}_{q_i}} \text{IsExcluded}(w) \right)$, where $\mathcal{W}_{q_i}$ is WCs for problem $q_i$, and $\text{IsExcluded}(w)$ is 1 if WC $w$ is eliminated, and 0 otherwise.

### 3.2 RESULTS

Table 1 presents the results for various model and method combinations on TC-Bench.

**TC-Bench Reveals a Significant Performance Ceiling for Current Technologies.** Even the best-performing combination, Claude-4 + HT, achieves less than 63%. This result strongly validates

Table 1: Performance comparison for all evaluated model-method combinations. **PR** denotes Pass-Rate and **HR** denotes HackRate. **AC** represents the percentage of non-excluded wrong codes. **WA**, **RE**, and **TLE** are all considered exclusions and contribute to HR. PSEUDO of Qwen3 is anomalous due to the API frequently returning empty or low-quality responses.

| LLM | Method | PR↑ | AC↓ | WA↑ | RE | TLE | HR↑ |
|---|---|---|---|---|---|---|---|
| **Open Source** | | | | | | | |
| Qwen2.5-32B | CRUX | 26.71 | 84.57 | 13.51 | 0.89 | 1.03 | 15.43 |
| | PSEUDO | 35.04 | 79.52 | 18.59 | 1.02 | 0.78 | 20.38 |
| | ALGO | 20.48 | 78.04 | 20.29 | 1.33 | 0.33 | 21.96 |
| | LCB | 57.62 | **48.39** | **48.46** | **2.07** | **1.08** | **51.61** |
| | HT | **65.46** | 69.27 | 29.17 | 1.22 | 0.34 | 30.73 |
| Qwen2.5-Coder-32B | CRUX | 22.68 | 81.27 | 16.31 | 0.91 | **1.51** | 18.73 |
| | PSEUDO | 37.72 | 79.23 | 18.72 | 0.98 | 1.07 | 20.77 |
| | ALGO | 21.33 | 81.41 | 17.27 | 0.85 | 0.46 | 18.59 |
| | LCB | 59.65 | **41.90** | **54.90** | **2.21** | 0.98 | **58.10** |
| | HT | **66.53** | 56.24 | 40.98 | 1.98 | 0.80 | 43.76 |
| Deepseek-V3 | CRUX | 37.90 | 83.01 | 15.54 | 0.85 | 0.60 | 16.99 |
| | PSEUDO | 19.58 | 88.32 | 10.97 | 0.37 | 0.34 | 11.68 |
| | ALGO | 28.22 | 70.78 | 27.53 | 1.24 | 0.44 | 29.22 |
| | LCB | 46.58 | **41.17** | **55.68** | **2.06** | **1.08** | **58.83** |
| | HT | **63.51** | 50.58 | 46.34 | 2.05 | 1.03 | 49.42 |
| Qwen3-235B-A22B | CRUX | 26.30 | 69.10 | 27.14 | 1.76 | 2.00 | 30.90 |
| | PSEUDO | 9.85 | 97.54 | 2.15 | 0.19 | 0.12 | 2.46 |
| | ALGO | 25.90 | 70.23 | 27.84 | 1.28 | 0.65 | 29.77 |
| | LCB | **70.40** | **54.03** | **41.25** | **2.40** | **2.32** | **45.97** |
| | HT | 55.35 | 69.20 | 28.50 | 1.61 | 0.69 | 30.80 |
| Qwen-Coder-Plus | CRUX | 29.26 | 67.65 | 28.79 | 1.71 | 1.85 | 32.35 |
| | PSEUDO | 40.15 | 67.11 | 29.57 | 1.45 | **1.87** | 32.89 |
| | ALGO | 30.43 | 67.04 | 31.15 | 1.31 | 0.50 | 32.96 |
| | LCB | **77.73** | **38.54** | **57.98** | **2.28** | 1.21 | **61.46** |
| | HT | 67.28 | 46.93 | 50.06 | 2.09 | 0.92 | 53.07 |
| **Closed Source** | | | | | | | |
| GPT-4o | CRUX | 42.43 | 70.77 | 26.25 | 1.46 | 1.52 | 29.23 |
| | PSEUDO | 50.90 | 73.33 | 24.01 | 1.03 | 1.63 | 26.67 |
| | ALGO | 24.43 | 75.51 | 22.87 | 0.97 | 0.65 | 24.49 |
| | LCB | **68.51** | **42.45** | **52.68** | **2.66** | **2.21** | **57.55** |
| | HT | 47.68 | 49.45 | 47.48 | 2.16 | 0.92 | 50.55 |
| Claude4 | CRUX | 32.93 | 76.31 | 21.11 | 1.14 | 1.44 | 23.69 |
| | PSEUDO | 64.72 | 63.97 | 32.35 | 1.23 | **2.45** | 36.03 |
| | ALGO | 32.12 | 69.17 | 29.01 | 1.20 | 0.62 | 30.83 |
| | LCB | 55.49 | 37.92 | 58.29 | 2.63 | 1.15 | 62.08 |
| | HT | **71.56** | **37.04** | **58.58** | **2.86** | 1.53 | **62.96** |
| Claude4-Thinking | CRUX | 30.47 | 66.26 | 31.14 | 1.44 | **1.16** | 33.74 |
| | PSEUDO | 23.56 | 85.98 | 12.84 | 0.51 | 0.67 | 14.02 |
| | ALGO | 32.41 | 64.54 | 33.68 | 1.22 | 0.56 | 35.46 |
| | LCB | **75.79** | **37.65** | **59.65** | 1.93 | 0.78 | **62.35** |
| | HT | 71.24 | 39.69 | 57.26 | **2.08** | 0.97 | 60.31 |

that WrongSelect indeed selects a diverse and challenging error basis, revealing a performance gap that would otherwise be masked in unfiltered benchmarks. This suggests that there is substantial room for improvement in handling complex and diverse errors, and TC-Bench serves as a reliable yardstick to measure this progress.

**A High PassRate does not Equate to a High Hackrate.** A high PassRate score can be hacked by generating a large number of easy test cases. For instance, on Qwen2.5-32B and Deepseek-V3, CRUX's PassRate is significantly higher than ALGO's, yet its Hackrate score is substantially lower.

**The Impact of Methodology Far Outweighs That of the Base Model.** The results consistently show that the choice of method has a much greater impact on final performance than the scale or even the source (open-source vs. closed-source) of the base model. For instance, while Qwen2.5-Coder-32B has fewer parameters than the activated parameters of Deepseek-V3, their HackRate scores with the LCB method differ by only 1%. In contrast, on Qwen2.5-Coder-32B, LCB's HackRate is

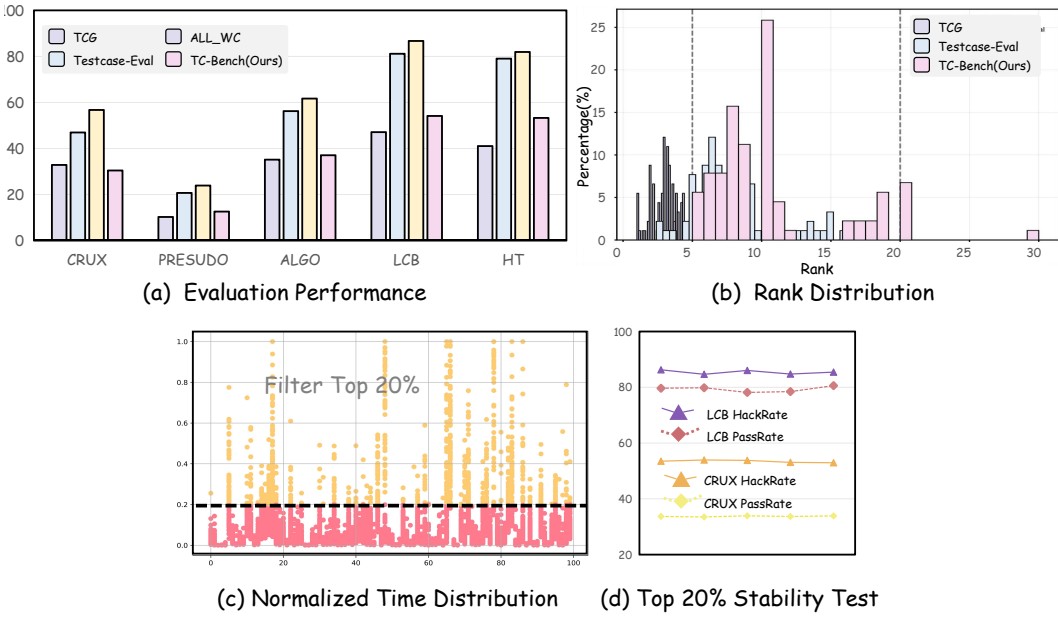

Figure 4: **(a)** Rank distribution of filtered WCs across methods. **(b)** Comparison of evaluation results against TC-BENCH (Ours). **(c)** Normalized execution times of correct solutions, primarily distributed below 0.2. **(d)** Sensitivity analysis showing stable PassRate and HackRate when filtering with random subsets of 8 correct solutions.

nearly 40% higher than CRUX. Furthermore, we observe that top-performing open-source models (e.g., Qwen-Coder-Plus) are competitive with leading closed-source models (e.g., the Claude4 series) across various methods. We hypothesize that this is because test case generation is a specialized task that is underrepresented in existing large-scale code pre-training corpora, thus limiting the performance gains through model scaling or a different training corpus. Further experimental analyses, study on Test-Time Scaling, and a summary of common error patterns, are detailed in Appendix A.3.

## 4 DISCUSSION

**Unfiltered and Heuristic Code Sets Lead to Biased Evaluation.** To validate the impact of code selection strategies, we conduct a rigorous comparison on a subset of 100 problems on Claude-4-Thinking. We compare TC-Bench against three baselines: TCGBench Ma et al. (2025b) (denoted as All_WC), which uses the full set of wrong codes; TestCase-Eval Cao et al. (2025), which randomly samples 20 codes; and TCG Yang et al. (2025), which selects 5 wrong codes from those passing at least 60% of test cases. **Unfiltered sets lead to score inflation.** As shown in Figure 4 (a), the full set leads to severe score inflation. For instance, LCB exhibits near-perfect performance ($\approx 100\%$) on All_WC, whereas its score on TC-Bench drops to just over 50%. This inflation masks the method's incompetence on core, difficult error patterns. Crucially, TestCase-Eval exhibits scores and trends highly similar to All_WC across all methods. This indicates that naive random sampling, while potentially reducing dataset size, fails to exclude redundant error patterns and thus cannot resolve the issue of score inflation. **Heuristic Selection results in under-representation.** Conversely, TCG yields significantly lower scores. While this might seem rigorous, our rank analysis reveals it stems from insufficient coverage. In Figure 4 (b), the rank of the error space varies significantly per problem. While most are below 20, some approach 30. TCG's rigid limit of 5 codes forces a drastic under-representation for high-complexity problems. While this lowers the performance scores of current methods, it makes future methods prone to score inflation: they would only need to cover a maximum subset of five patterns rather than the complete error space to achieve perfect scores. In summary, TC-Bench strikes the optimal balance. By selecting a basis defined by the problem's

intrinsic rank, it avoids both the inflation of coverage-based methods and the under-representation of heuristic constraint methods, serving as a stable and fair test suite.

**Rank serves as the Upper Bound for the Necessary Number of Test Cases.** The row rank, which represents the number of independent error patterns, equals the column rank, which represents the number of independent diagnostic dimensions. In an error space defined by rank $R$, there are only $R$ linearly independent diagnostic dimensions. Any additional test case is merely a linear combination of these basis dimensions and does not provide new information for distinguishing existing error patterns. Therefore, $R$ test cases are sufficient to distinguish all error patterns, serving as a compact upper bound. Consider a concrete example matrix with $R = 3$:

|       | $t_1$ | $t_2$ | $t_3$ | $t_4$ |
|-------|-------|-------|-------|-------|
| $w_1$ | 1     | 0     | 1     | 0     |
| $w_2$ | 0     | 1     | 1     | 0     |
| $w_3$ | 0     | 1     | 1     | 0     |
| $w_4$ | 0     | 0     | 0     | 1     |

Here, columns $t_1$ and $t_2$ are linearly independent, but $t_3$ is a linear combination ($t_3 = t_1 + t_2$). Any wrong code failing on $t_1$ or $t_2$ implies a predictable behavior on $t_3$. Thus, $t_3$ offers no new diagnostic dimension. The set $\{t_1, t_2, t_4\}$ is sufficient to distinguish all unique error patterns. This framework addresses a critical flaw in previous evaluations where the number of test cases was arbitrary. Consequently, problems with small diagnostic dimensions were often "over-tested," inflating scores, while complex problems were "under-tested." Using Rank as the budget ensures fairness: it allows simple problems to reveal performance gaps while ensuring complex problems are tested with sufficient depth.

**Correct Code Selection Influence Results.** Unlike WCs, which have failure signatures, correct codes all behave identically on GT, differing only in runtime and memory usage. This makes their selection more subtle. Using only a single correct solution as a validator is insufficient. Certain invalid input may still have an output under a specific code. Our initial exploration shows that as the number of correct codes increases (as shown in Figure 8, more ATs are filtered, leading to higher HackRates. However, not all filtering is beneficial. Many complex but valid ATs are wrongly discarded due to timeouts by slow correct codes. Worse, such low-performance correct codes show inconsistency across environments (different OJ platforms). Performance profiling reveals a highly skewed distribution: most correct codes cluster in the top 20% after applying min–max normalization to runtimes (Figure 4 (c)). These high-performance codes are stable across platforms. Consequently, we adopt a biased random sampling strategy: for each problem, we retain only correct codes within the top 20% normalized runtime and randomly sample 8 from this set. Repeated experiments confirm that this strategy yields highly stable evaluation outcomes ( Figure 4 (d)).

**AT Uncover Latent Bugs Beyond GT.** An interesting phenomenon emerged during evaluation: some wrong codes labeled as Wrong Answer under GTs produce Runtime Error or Time Limit Error when executed on ATs. To verify whether this is due to server overload, we conduct a controlled experiment. We sample 350 WCs that exhibited RE/TLE and combined them with about 2.6k random WCs. Running these on a 128-core machine, we gradually reduce concurrency from 128 to 88 tasks. The RE/TLE frequency remains nearly constant regardless of system load. This strongly suggests that advanced methods are indeed capable of producing stricter and more challenging ATs than official GTs, revealing hidden bugs related to performance and robustness.

## 5 CONCLUSION

Existing evaluation practices suffer from inflated scores and unclear principles regarding how many codes and test cases are necessary. We addressed this gap by formalizing benchmark construction as a binary-matrix rank problem, which jointly determines the minimal code basis and the upper bound on test cases. To approximate its NP-hard solution, we introduced WrongSelect and applied it to large-scale competitive programming data, resulting in TC-Bench, a compact and diverse diagnostic benchmark. Experiments show that TC-Bench reveals substantial gaps in current methods and provides a faithful foundation for advancing research on test case generation.

ETHICAL STATEMENT

The data for the proposed methods is drawn solely from publicly accessible project resources on reputable websites, ensuring that no sensitive information is included. Moreover, all datasets and baseline models used in our experiments are also available to the public. We have taken care to acknowledge the original authors by properly citing their work.

ACKNOWLEDGE

We gratefully acknowledge the support of the National Natural Science Foundation of China (NSFC) via grant 62236004 and 62476073.

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

# A  APPENDIX

## A.1  RELATED WORK

**Test Case Generation**   As private ground-truth test cases are scarce, researchers have turned to LLMs for automatic test case generation.(Cook et al., 2025; Chen et al., 2025; Shi et al., 2025; Seed et al., 2025; Fatemi et al., 2025; Ahmed et al., 2024; Yu et al., 2025b; Zhoubian et al., 2025; Lei et al., 2024) Early work had models directly produce complete test cases, i.e., input-output pairs.(Gu et al., 2024; Chen et al., 2023; Zeng et al., 2025b; Xu et al., 2025b; Payoungkhamdee et al., 2025), However, because such outputs are often unreliable,  Jiao et al. (2024); Li et al. (2023) let the model generate both an input and a candidate solution, then execute the solution to derive the output. Other methods introduced input generators to replace raw inputs (Jain et al., 2024; Cao et al., 2025; Xia et al., 2025), or validators to enforce format and range constraints before execution (He et al., 2025; Fu et al., 2025a). Some methods enhance the model's ability to generate test cases through training, such as via SFT (Supervised Fine-Tuning), RL(Reinforcement Learning)and other techniques. (Li et al., 2025; Bai et al., 2025; Zhang et al., 2024; Wang et al., 2025a) Most recently, multi-round generation and execution feedback has led test case generation to agent workflows (Wang et al., 2025c; Da et al., 2025; Ye et al., 2025; Zhang et al., 2025; Yu et al., 2025a; Huang et al., 2024).

**Test Case Evaluation**   Evaluation originally followed traditional software testing, emphasizing coverage and distinguishing between buggy and fixed code. (Xu et al., 2025a; Yu et al., 2025c) SWT-Bench (Mündler et al., 2025) and TestGenEval (Jain et al., 2025) transform from SWE-Bench (Jimenez et al., 2024), providing buggy implementations and their corresponding fixes. Others extend beyond single languages or update to recent codebases. For algorithmic problems, TestE-val collected 210 problems but still relied on coverage metrics. More recent works shifted toward end-to-end evaluation with large collections of correct and wrong submissions, measuring how often generated test cases exclude incorrect code. (Ma et al., 2025a; Yang et al., 2025; Wang et al., 2025b) However, these approaches either rely on ad-hoc manual selection or expand code sets without selection or analysis. TC-Bench is the first to study how many codes and test cases are sufficient, and provides a principled, efficient evaluation framework.

**Code-Test Matrix**   CodeT Chen et al. (2023) and B4 Chen et al. (2024) share the concept of using execution results on test cases (the Code-Test Matrix) as behavioral signatures. CodeT assumes correct code behaviors are consistent while incorrect results are diverse, utilizing signatures for clustering to select a consensus set. B4 calculates the probability of observing the matrix to select the most likely correct cluster. However, these methods aim for solution selection where the correctness of code and tests is unknown, utilizing the matrix primarily for signature matching or probabilistic modeling. In their context, the algebraic rank and basis are not the primary interpretative tools. In contrast, TC-Bench operates on ground truth with guaranteed correct tests and wrong codes. We view the matrix as a complete Error Space and apply linear algebra operations to calculate the Rank and Basis, representing this error space most efficiently.

## A.2 WRONGSELECT

---

**Algorithm 1** WrongSelect

---

1: **Input:** Raw matrix $\mathbf{M}$, filter threshold $\tau$, restart count $E$, local search step $K$
2: **Output:** The optimal basis $\mathcal{I}^*$
3:                                                             ▷ *Phase 1: Principled Pre-filtering*
4: $\mathbf{M}' \leftarrow \text{Filter}(\mathbf{M}, \tau)$
5: $R' \leftarrow \text{rank}(\mathbf{M}')$
6: $\mathcal{I}^* \leftarrow \emptyset$
7: $F_{\min} \leftarrow \infty$
8:                                              ▷ *Phase 2: Random-Restart Local Search*
9: **for** $i = 1$ to $E$ **do**
10:     $\mathcal{I}_{\text{current}} \leftarrow \text{RandomBasis}(\mathbf{M}', R')$                  ▷ Generate a random initial basis
11:     $F_{\text{current}} \leftarrow F(\mathcal{I}_{\text{current}})$
12:     **for** $j = 1$ to $K$ **do**
13:         $\mathcal{I}_{\text{best\_neighbor}} \leftarrow \mathcal{I}_{\text{current}}$
14:         $F_{\text{best\_neighbor}} \leftarrow F_{\text{current}}$
15:         **for** each $\mathbf{r}_{\text{in}} \in \mathbf{M}' \setminus \mathcal{I}_{\text{current}}$ and each $\mathbf{r}_{\text{out}} \in \mathcal{I}_{\text{current}}$ **do**
16:             $\mathcal{I}_{\text{temp}} \leftarrow (\mathcal{I}_{\text{current}} \setminus \{\mathbf{r}_{\text{out}}\}) \cup \{\mathbf{r}_{\text{in}}\}$              ▷ Traverse each neighbor
17:             **if** $\text{rank}(\mathcal{I}_{\text{temp}}) = R'$ **then**
18:                 **if** $F(\mathcal{I}_{\text{temp}}) < F_{\text{best\_neighbor}}$ **then**
19:                     $\mathcal{I}_{\text{best\_neighbor}} \leftarrow \mathcal{I}_{\text{temp}}$
20:                     $F_{\text{best\_neighbor}} \leftarrow F(\mathcal{I}_{\text{temp}})$
21:                 **end if**
22:             **end if**
23:         **end for**
24:         **if** $F_{\text{best\_neighbor}} < F_{\text{current}}$ **then**               ▷ Move to the best neighbor
25:             $\mathcal{I}_{\text{current}} \leftarrow \mathcal{I}_{\text{best\_neighbor}}$
26:             $F_{\text{current}} \leftarrow F_{\text{best\_neighbor}}$
27:         **else**
28:             **break**                  ▷ Local optimum reached, exit inner loop
29:         **end if**
30:     **end for**
31:     **if** $F_{\text{current}} < F_{\min}$ **then**
32:         $F_{\min} \leftarrow F_{\text{current}}$
33:         $\mathcal{I}^* \leftarrow \mathcal{I}_{\text{current}}$
34:     **end if**
35: **end for**
36:
37: **return** $\mathcal{I}^*$

---

Phase 2 in Algorithm 1 illustrates the pseudo code. The algorithm consists of two nested loops: the outer loop explores multiple random starting points to ensure global search breadth, while the inner loop refines each starting point to a local optimum, ensuring local search depth. Given the pre-filtered matrix $\mathbf{M}'$, each outer iteration begins by generating a random initial basis $\mathcal{I}$current. The inner loop then iteratively improves this basis. In each iteration, the algorithm systematically explores the neighborhood of the current basis: a neighbor basis is obtained by swapping one member inside the basis with one outside, while maintaining the same rank. We compute the average Jaccard similarity $F(\mathcal{I}$temp$)$ for each neighbor. If the best neighbor $\mathcal{I}_{\text{best\_neighbor}}$ is superior to the current solution, $\mathcal{I}$current is updated accordingly, and the process continues. Otherwise, when no better neighbor exists, the algorithm concludes that a local optimum has been reached and the inner loop terminates. After each outer iteration, the current basis is compared with the best basis found so far, and the best is updated if necessary. The outer loop repeats this procedure from multiple random initializations, and finally, the best basis across all runs is returned as the approximate global optimum.

Table 2: Model Performance Comparison

| LLM | Method | PR ↗ | AC ↘ | WA ↗ | RE | TLE | HR ↗ |
|---|---|---|---|---|---|---|---|
| Qwen2.5-7B | Crux | 26.86 | 81.44 | 16.14 | 0.89 | 1.53 | 18.56 |
| | PSEUDO | 9.52 | 98.86 | 1.06 | 0.05 | 0.04 | 1.14 |
| | ALGO | 12.37 | 89.61 | 9.25 | 0.67 | 0.46 | 10.39 |
| | LCB | 42.38 | **52.46** | **43.69** | **2.29** | **1.56** | **47.54** |
| | HT | **58.51** | 68.78 | 28.66 | 1.53 | 1.03 | 31.22 |
| Qwen2.5-14B | Crux | 29.12 | 81.22 | 16.36 | 1.00 | 1.42 | 18.78 |
| | PSEUDO | 14.97 | 93.92 | 5.30 | 0.32 | 0.46 | 6.08 |
| | ALGO | 19.82 | 86.91 | 11.82 | 0.71 | 0.56 | 13.09 |
| | LCB | 49.71 | **49.65** | **46.63** | **2.17** | **1.55** | **50.35** |
| | HT | **70.79** | 64.23 | 33.83 | 1.29 | 0.64 | 35.77 |
| Qwen2.5-Coder-7B | Crux | 33.13 | 80.53 | 17.15 | 1.15 | 1.18 | 19.47 |
| | PSEUDO | 16.49 | 88.65 | 10.31 | 0.55 | 0.50 | 11.35 |
| | ALGO | 14.27 | 92.80 | 6.62 | 0.40 | 0.17 | 7.20 |
| | LCB | 41.94 | **57.83** | **39.02** | **1.92** | **1.23** | **42.17** |
| | HT | **71.02** | 78.08 | 20.47 | 0.97 | 0.48 | 21.92 |
| Qwen2.5-Coder-14B | Crux | 26.18 | 81.27 | 16.44 | 1.05 | 1.24 | 18.73 |
| | PSEUDO | 10.32 | 95.46 | 4.16 | 0.28 | 0.10 | 4.54 |
| | ALGO | 27.92 | 95.45 | 4.20 | 0.22 | 0.13 | 4.55 |
| | LCB | 51.87 | **46.05** | **49.95** | **2.34** | **1.66** | **53.95** |
| | HT | **73.07** | 68.45 | 29.75 | 1.21 | 0.58 | 31.55 |

## A.3 MAIN RESULTS

**The Usage of Correct Code is a Performance Watershed.** Across nearly all models, methods that rely on correct code (LCB, HT) significantly outperform those that do not (CRUX, PSEUDO, ALGO) on HackRate. Although methods like PSEUDO and ALGO attempt to ensure correctness by having the LLM generate its own solution (or even a simpler brute-force one), the success of this process is constrained by the LLMs' own problem-solving capabilities. When the model generates an incorrect solution, it not only fails to generate complex test cases, but even simple ones are filtered out due to incorrect outputs. All this leads to a low PassRate, which in turn severely impacts the Hackrate. Their performance is sometimes even worse than the simplest CRUX method.

**Performance Gains Primarily Come From WA.** Through a fine-grained analysis of exclusion reasons, we find that the primary performance gain from advanced methods with specific edge case generators, such as LCB and HT, comes from a significantly improved WA exclusion rate. For error types like RE and TLE, scores do not show a significant gap compared to simpler methods like CRUX. This suggests that the core advantage of current SOTA methods lies in generating ingenious test cases that probe for algorithmic logic flaws. Crafting test cases that effectively trigger robustness failures may be a different, and perhaps a more difficult challenge.

**Implementation Details Significantly Impact Final Performance.** Although the five methods are conceptually progressive, specific implementation details, such as prompts and pipelines, can cause substantial performance variations. The concepts of ALGO and PSEUDO are similar, but ALGO simplifies the task by asking the model to generate a simpler brute-force solution. However, PSEUDO often outperforms ALGO because it generates 10 solutions and uses a majority vote, whereas ALGO generates only one. Similarly, although HT adds an input validator over LCB, it underperforms on most models. We attribute this to implementation choices, such as allowing the edge case generator to return empty and providing simpler few-shot examples, which may lead the model to "get lazy" and produce less complex test cases.

## A.4 TEST TIME SCALING

To investigate the quantitative impact of increasing the number of test cases, we conduct a scaling experiment. For each problem, we used its rank $R'$ as the base number of test cases (1x) and proportionally scaled this number up to 5x, observing the trend in HackRate.

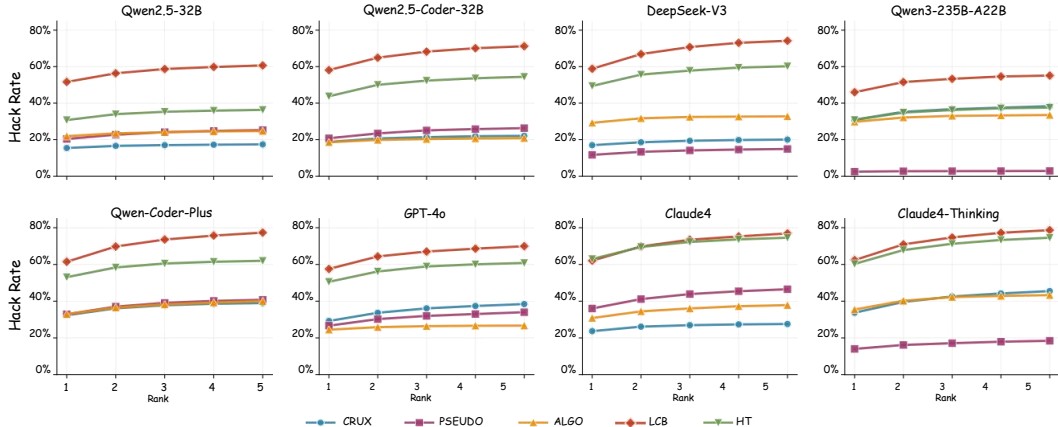

Figure 5: Results of test case scaling for each model and method. The x-axis represents the number of test cases, scaled as multiples of the problem's rank from 1x to 5x.

**The addition of test cases exhibits significant diminishing returns.** As shown in Figure 5, the gain from scaling from 1x to 2x is the most significant across all combinations. As the number increases from 3x to 5x, the performance curves generally begin to flatten, or even saturate. This suggests that blindly and massively increasing the number of test cases is an inefficient strategy. After covering the regular error patterns, additional test cases are likely just re-validating known failures rather than uncovering new, deeper defects.

**The relative performance ranking among methods remains highly stable across all scales.** Crucially, this experiment validates the effectiveness of setting the base number of test case as the problem's rank $R'$. While increasing the number of test cases does improve HackRate, the performance curves for each method almost never intersect. For instance, for Deepseek-V3 and GPT-4o, the five methods are well-separated. This stability demonstrates that TC-Bench, is already an efficient and reliable benchmark for differentiating the performance of various test case generation methods. It successfully captures the core discriminative power of different methods without incurring the high computational cost of scaling.

The core conclusions from our main experiments demonstrate good scale-invariance. Finally, this scaling experiment further reinforces the core findings from our main experiments. For example, the performance gap between methods that rely on correct code (LCB, HT) and those that do not remains significant at all test case scales. Similarly, the impact of methodology continues to outweigh that of the base model.

## A.5 COMMON FAILURE

To better understand the causes behind low scores, we conducted a qualitative analysis of failed generations and identified three major systematic shortcomings.

**Task Confusion and Instruction-Following Failures** When prompted to generate test cases, many LLMs instead output complete solutions to the problem. This issue is particularly common when both test cases and solutions are requested together. We hypothesize that this stems from the infrequency of test-case generation tasks in training data and weakened instruction-following ability after long-cot training (Fu et al., 2025b). DeepSeek-R1 exhibited this issue most severely. As shown in Figure 6, within CRUX and PSEUDO, 74% and 60% of its outputs, respectively, are direct solution code rather than valid test cases. Among the remaining outputs, many are unusable due to formatting errors, such as embedding executable code inside JSON. Because the extractable test cases are too few, R1 is excluded from the main experiments. This finding highlights that successful test-case generation requires not only strong reasoning ability, but also precise task comprehension and robust formatting control.

```json
{
  "test_case_0": "7\n1 1 2 1 4 6\n5\n3 4 6\n1
5 7\n1 2 4\n1 2 6\n1 3 5",
```
✓

( a ) Standard CRUX Response

```
"test_case_9": "300000\n" + "
".join(str(max(1, i-10)) for i in range(1,
300000)) + "\n300000\n" + "\n".join(f"1 {i}
{i+2}"for i in range(2, 300002))"
```
✗

```
arr =[0] *n_val
arr_str =" ".join(map(str, arr))
op_lines =[]
for j in range(n_val):
    op_lines.append(f"1 {n_val} {j}")
test_cases["test_case_5"]
=f"{n_val}\n{arr_str}\n"+"\n".join(op_lines)
```
✗

( b ) Generate Testcase-Code

**To solve this problem**, we need to determine the
number of players observed by each observer placed at
different nodes in a tree. Each player starts at their
respective starting node and moves along the shortest
path to their destination node at a speed of one edge
per second. An observer at node $( j )$ observes players
who arrive exactly at $( W_j )$ seconds. The solution
involves efficiently counting the players who pass
through each node $( j )$ at exactly $( W_j )$ seconds.
### Solution Code
```python
import sys
sys.setrecursionlimit(500000)

def main():
    data = sys.stdin.read().split()
    if not data:
        return
......
```
✗

Not TestCase but Solution Code

( c ) Direct Solution Code

Figure 6: The figure shows the errors that occur in the direct generation of Testcases by models like R1, using the CRUX and PRESUDO algorithms. Subfigure (a) shows the normal output, subfigure (b) demonstrates the insertion of generated code or direct responses in the form of Testcase code within a string, and subfigure (c) shows the model not following the Testcase generation instructions and instead directly providing the solution.

**Lack of Resource-Aware Generation**  Many problems require test cases at large boundary conditions. As shown in Figure 7 a, we observe that numerous methods attempt to construct overly large inputs (e.g., huge graph structures), leading to out-of-memory crashes or timeouts during execution. This reveals a deeper limitation: while LLMs are proficient in generating algorithmic logic, they lack awareness of physical execution constraints such as memory and runtime. A robust test-case generation pipeline must therefore incorporate mechanisms like input partitioning or streaming to adapt to limited system resources.

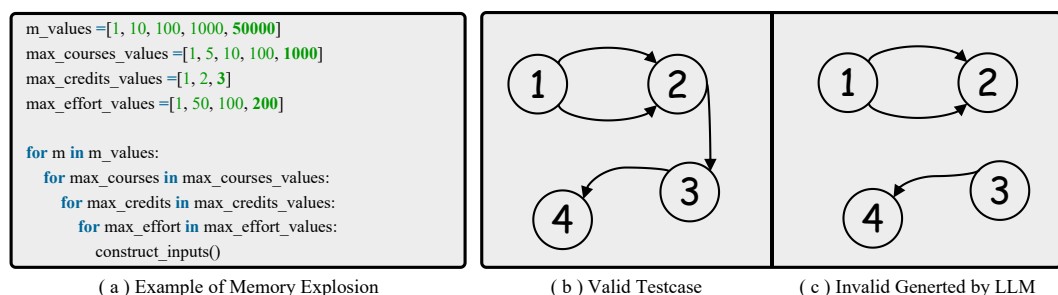

```
m_values =[1, 10, 100, 1000, 50000]
max_courses_values =[1, 5, 10, 100, 1000]
max_credits_values =[1, 2, 3]
max_effort_values =[1, 50, 100, 200]

for m in m_values:
    for max_courses in max_courses_values:
        for max_credits in max_credits_values:
            for max_effort in max_effort_values:
                construct_inputs()
```

( a ) Example of Memory Explosion       ( b ) Valid Testcase       ( c ) Invalid Generted by LLM

Figure 7: Subfigure a shows that the memory explosion is caused by the model constructing an excessively large number of functions during case generation. Subfigure b, c presents a failed case where the model fails to construct a connected graph as required. The task specifies that all node 1 instances must be able to reach node n, but the constructed graph does not satisfy this connectivity condition.

**Failure to Construct Required Complex Data Structures**   Some problems in our benchmark admit only test cases with highly constrained structures. As shown in Figure 7 b, c, in one problem, every valid input must be a specific type of connected graph. However, none of the tested methods successfully produced even a single valid input. As a result, these problems ended up with zero usable test cases. This underscores that generating high-difficulty test cases can be as challenging as solving an algorithmic problem, requiring a deep understanding of both data structures and algorithms.

## A.6   RESULTS OF SUPPLEMENTARY EXPERIMENTS

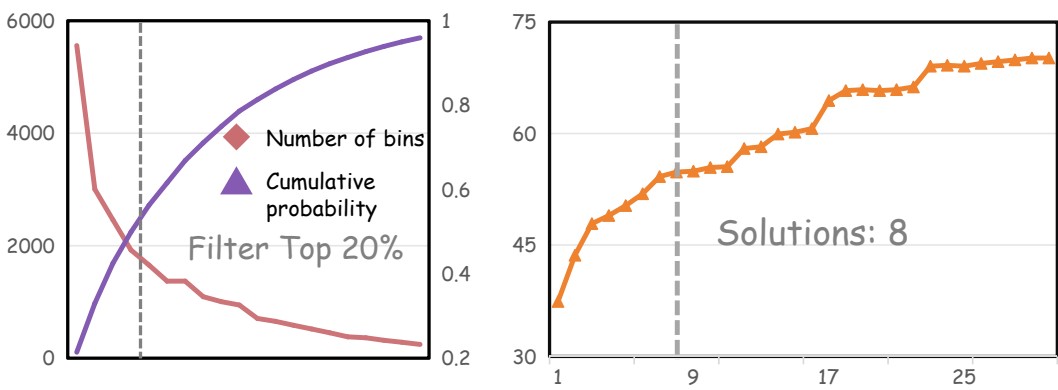

Figure 8: The left subplot shows the interval count statistics and cumulative probability curve of the time-normalized correct answers. In the right subplot, the Hackrate continuously increases as the number of correct answers increases.

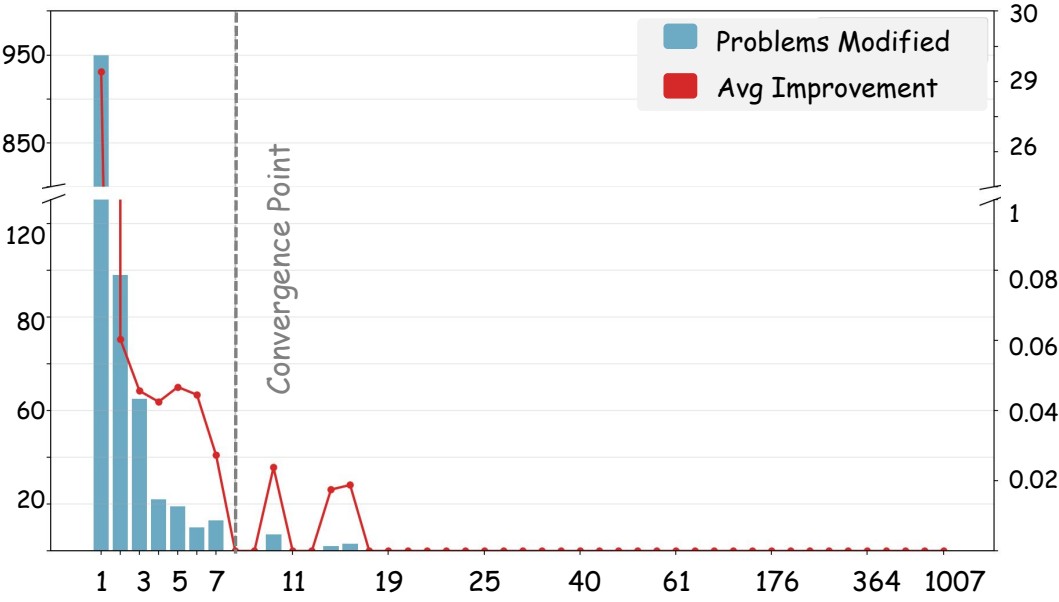

Figure 9: The heuristic algorithm we designed converges at the eighth turn.

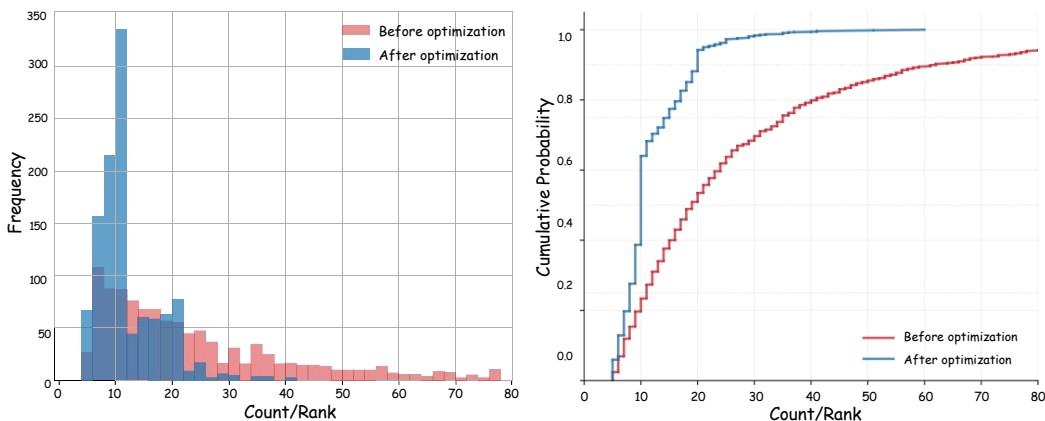

Figure 10: Distribution of the number of WCs per problem before and after the WrongSelect. The histogram (left) compares the initial count of WCs against the rank (i.e., the final count of WCs). The cumulative distribution function (CDF) on the right further illustrates this shift. The results demonstrate a dramatic reduction in the number of required codes, highlighting the compactness and efficiency of our resulting benchmark.

## A.7 THE USE OF LARGE LANGUAGE MODELS (LLMS)

This article utilizes large language models (LLM) solely for writing refinement and graphic enhancement, with no other applications or purposes involved.

# B  APPENDIX B

## B.1  BENCHMARK CONSTRUCTION

This section will detail the process involved in constructing the dataset, including the repairing of Wrong Codes, operations related to the clarity of problem statements, and statistical data.

### B.1.1  WRONG CODE

**Code Cleaning**    After processing the wrong codes in Section2.3, for all retained wrong codes, we used public test cases for testing. For all execution results such as CE, TLE, MLE, EXE, as well as codes that resulted in WA but with empty outputs, manual fixes and reviews were performed. As shown in Figure 11, (a) illustrates a piece of unusable file operation code, in which the script does not include the definitions of Fin and Fout. For this type of code, we removed the corresponding file operations. The error in (b) arises because the unistd.h library already defines a function named *link_array*, which conflicts with the array *link_array* defined in the code. (c) presents an example of incomplete code that requires manual supplementation. To ensure consistency between the original and the corrected code, after making modifications we tested the code using private test cases, with the requirement that the test results remain consistent with the crawled results.

```
il void FILEIO(){
    #ifdef intLSY
        Fin("in.in");
    #endif
}
il void FILEIO( string pname ){
    #ifndef intLSY
        Fin((pname+".in").c_str());
        Fout((pname+".out").c_str());
    #endif
}
il void FILEIO_OICONTEST( string pname ){
    Fin((pname+".in").c_str());
    #ifndef intLSY
        Fout((pname+".out").c_str());
    #endif
}
```

Undeclared function: Fin, Fout

( a ) Unavailable file operation

```
# include<unistd.h>

int ch[maxn<<1][26], link_array[maxn<<1]

extern intlink_array(intoldfd, const char *oldpath,
    intnewfd, const char *newpath, intflags);
```

( b )Standard library name conflict

```
int main()
{
    n=qread(),m=qread();
    for(int i =1; i <=n; i++)
    {
        int cnt;
        cnt=qread();
```

Manually complete the code

( c ) Incomplete code

Figure 11: (a), (b), and (c) respectively present three examples that we encountered when repairing Wrong Code.

### B.1.2 PROBLEM DESCRIPTION

Regarding the problem statement processing in Section2.3, this subsection provides detailed examples and explanations for problems that heavily rely on images and Special Judge problems.

For problem statements that rely on image-based understanding, such as Stars (see image in reference Figure12), the problem includes an image that is necessary for understanding in order to generate test cases or solve the problem. We manually reviewed this type of problem statement, filtered out the problems where images affected the understanding of the question, and deleted them. In this step, we deleted a total of 71 problems.

Star

There are some stars in the sky, each with a different position, and each star has a coordinate. If a star has k stars in its lower-left (including directly left and directly below), we say that this star is of level k.

For example, in the image below, star 5 is of level 3 (because stars 1, 2, and 4 are in its lower-left), and stars 2 and 4 are of level 1. In the example image, there is 1 star of level 0, 2 stars of level 1, 1 star of level 2, and 1 star of level 3.

Given the positions of the stars, output the count of stars at each level.

Figure 12: This is an example of a problem that can only be solved with image understanding.

The problems with Special Judges involve multiple outputs, answer ranges, and interactive problems. In total, we removed 42 Special Judge problems. Ball Moving Game is an example with multiple solutions, as shown clearly in Figure 13, which illustrates the existence of multiple answers. Problems like Idea also explain that as long as the answer satisfies a certain range, it is acceptable.

Ball Moving Game：

Little C is stuck, but he believes you can solve it. Please provide an operational plan to achieve Little C's goal. There may be multiple valid solutions, and you only need to provide one. The problem guarantees that there is always at least one valid solution.

Idea:

For each output file, if more than 95% of the lines have an answer with an error of no more than 25% compared to the correct answer, you will receive a score. The error is considered to be within 25% if, for a correct answer X, your answer lies within the closed interval [0.8X, 1.25X] .

Figure 13: The image presents two examples of problems with multiple solutions. In the Ball Moving Game, the same input can lead to various outputs, while in the Idea problem, the output simply needs to fall within a given range.

We also selected all interactive problems, such as the one shown in the reference, The Adventure of Lord I, where the problem statement clearly states "*This is an interactive problem.*" This type of problem requires complex interactions and support, making it unfriendly for test case generation.

---

The Adventure of Lord I:

This is an interactive problem.
During the evaluation, the interactive library will call the `explore` function exactly once.

It is guaranteed that the graph used in this problem is fully determined before the interaction begins and will not be dynamically constructed based on the interactions with your program. Therefore, the interactive operations in the problem are deterministic, and you do not need to worry about the specific implementation of these operations in the interactive library.
The data guarantees that the time required for the interactive library to run will not exceed 1 second under the given call limits. The memory used by the interactive library is fixed and does not exceed 128MB.

---

Figure 14: The image illustrates an example of an interactive problem, which necessitates specific and intricate interaction checks during evaluation. To streamline the evaluation process, we have removed this part of the problem.

**Example of problem statement cleaning**    As shown in Figure15,the following is an example of problem statement cleaning. For demonstration purposes, we have created a sample problem to illustrate the main cleaning tasks. In this process, irrelevant background information is removed, image links and other URLs are discarded, and the phrasing is made smoother. The data range is kept to the most general case. These tasks typically do not follow a universal pattern and require manual inspection. After cleaning the problem statement, we used GPT-4o for translation. In this step, we organized each data entry and deleted 15 problems that were difficult to handle. Then each translation was semantically proofread and certain inappropriate expressions were adjusted for accuracy.The final processed problem statement can be found in next page

Tour de Byteotia

1 Markdown Format

# Tour de Byteotia

Background:

~~In the depths of a distant universe, there exists a kingdom surrounded by stars and brilliance—"The Kingdom of Stars." This kingdom is home to countless magical scholars who explore mysterious stellar trajectories and intertwined fates. One day, the scholars discovered an ancient prophecy foretelling that a broken constellation would bring about the end of the world. Only by gathering five lost stellar gems can this disaster be prevented. And you, the chosen hero, bear the heavy responsibility of changing fate.~~

Problem Description:

2 Remove Irrelevant Message

Given an undirected graph with n vertices and m edges, determine the minimum number of edges to remove so that vertices numbered less than or equal to k do not appear on any simple cycle.

Input Format:

The first line contains three integers n , m , and k , representing n nodes, m edges, and k as described in the problem statement.

Output Format:

The first line contains one integer representing the minimum number of edges to be removed;

The following lines each output two positive integers a,b , representing the removal of the edge between a and b . Output the vertex with the smaller number first, then the vertex with the larger number.

Sample:

3 ADD Sample

Input:

Output:

4 Remove Fig, URL , HTML

~~~~

Constraints and Hints:

5 General Description

~~For 40% of the data, n≤1000,m≤5000 .~~

For all data, $1 \leq n \leq 1,000,000, 0 \leq m \leq 2,000,000, 1 \leq k \leq n, 1 \leq u < v \leq n$ .

Figure 15: To facilitate demonstration, we constructed an example of problem-statement cleaning, in which the common cleaning procedures are integrated.

**Final problem statement:**

```
# Tour de Byteotia

## Problem Description

Given an undirected graph with $n$ vertices and $m$ edges,
    ↪ determine the minimum number of edges that need to be removed
    ↪  so that all vertices with indices less than or equal to $k$
    ↪ are not part of any simple cycle.

## Input Format

The first line contains three integers $n$, $m$, and $k$,
    ↪ representing the number of vertices, the number of edges, and
    ↪  the significance of $k$ as described in the problem
    ↪ statement.

The next $m$ lines each contain two integers $u$ and $v$,
    ↪ indicating a bidirectional edge between $u$ and $v$. There is
    ↪  at most one edge between any pair of vertices.

## Output Format

The first line contains an integer $k$, representing the minimum
    ↪ number of edges to be removed.

The next $k$ lines each contain two positive integers $a$ and $b$,
    ↪ indicating the removal of an edge between $a$ and $b$. Output
    ↪  the vertex with the smaller index first, followed by the
    ↪ vertex with the larger index.

## Examples

### Input:
11 13 5
1 2
1 3
1 5
3 5
2 8
4 11
7 11
6 10
6 9
2 3
8 9
5 9
9 10

### Output:
3
2 3
5 9
3 5

## Data Range and Hints
For all data, $1 \le n \le 1,000,000$, $0 \le m \le 2,000,000$, $1
    ↪ \le k \le n$, $1 \le u < v \le n$.
```

## C  CASE STUDY

To demonstrate the practical effectiveness of our method, we conduct a case study on "Sliding Window", a classic problem requiring the Monotonic Queue algorithm. The problem involves an integer array of length $N(\leq 10^6)$ and a window of size $K(\leq 10^6)$. The window slides from the leftmost to the rightmost of the array, moving one position at a time. The goal is to determine the maximum and minimum values within the window at each step. The output requires two lines: the sequence of minimums followed by the sequence of maximums.

The optimal solution employs a Monotonic Queue to achieve a time complexity of $O(N)$. Specifically, to calculate the maximums, we maintain a monotonically decreasing queue that stores array indices. As we iterate through each element in the array, we first pop the elements at the back of the queue if their corresponding values are less than or equal to the current element. This is because these smaller and older elements can never serve as the maximum for future windows. Next, the current index is pushed to the back. Then, the front of the queue is popped if its index is out of the current window scope. Finally, the value corresponding to the index at the front of the queue represents the maximum of the current window. The minimums are calculated analogously by maintaining a monotonically increasing queue. The standard solution is shown below.

---

**Standard Solution**

```cpp
#include<bits/stdc++.h>
using namespace std;
int n , a[1000005] , k ;
deque<int>q ;
int main() {
  cin >> n >> k ;
  for (int i = 1 ; i <= n ; i ++) {
    cin >> a[i] ;
  }
  for (int i = 1 ; i <= n ; i ++) {
    while(!q.empty() && a[i] < a[q.back()]) {
      q.pop_back() ;
    }
    q.push_back(i) ;
    if(q.front() < i - k + 1) {
      q.pop_front() ;
    }
    if(i >= k) cout << a[q.front()] << " " ;
  }
  cout << endl ;
  q.clear() ;
  for (int i = 1 ; i <= n ; i ++) {
    while(!q.empty() && a[i] > a[q.back()]) {
      q.pop_back() ;
    }
    q.push_back(i) ;
    if(q.front() < i - k + 1) {
      q.pop_front() ;
    }
    if(i >= k) cout << a[q.front()] << " " ;
  }
}
```

---

Initially, this problem involves 96 Wrong Codes (WCs). After applying WrongSelect, only 8 basic WCs are retained. Their failure signatures are presented below:

$$
\mathcal{I}^* = \begin{bmatrix}
0 & 0 & 0 & 0 & 0 & 0 & 0 & 1 & 1 & 0 \\
1 & 1 & 1 & 1 & 0 & 0 & 1 & 1 & 1 & 1 \\
0 & 0 & 0 & 0 & 0 & 0 & 1 & 0 & 0 & 0 \\
0 & 0 & 0 & 1 & 0 & 0 & 1 & 0 & 1 & 1 \\
0 & 0 & 0 & 1 & 0 & 0 & 0 & 0 & 1 & 0 \\
0 & 0 & 0 & 0 & 0 & 1 & 0 & 1 & 0 & 1 \\
0 & 0 & 0 & 0 & 0 & 0 & 0 & 1 & 0 & 0 \\
1 & 1 & 0 & 0 & 0 & 0 & 0 & 0 & 0 & 0
\end{bmatrix}
$$

## C.1 BASIC WRONG CODES

We meticulously analyze the retained Basic WCs and characterize their underlying error patterns.

Basic WC1 fails due to insufficient memory allocation for the queue array, where the size is set to 510,000 instead of the required 1,000,010.

**Basic WC1 (0000000110)**

```cpp
#include<cstdio>
#include<cstring>
using namespace std;
struct node {
  int x,p;
}
list1[510000],list2[510000];// Should expand 510000 to 1010000
int a[510000],n,m;
int main() {
  scanf("%d%d",&n,&m);
  for (int i=1;i<=n;i++) scanf("%d",&a[i]);
  int head=1,tail=1;
  list1[1].x=a[1];
  list1[1].p=1;
  for (int i=2;i<=n;i++) {
    while(head<=tail&&i-list1[head].p>=m)head++;
    while(head<=tail&&list1[tail].x>=a[i])tail--;
    list1[++tail].x=a[i],list1[tail].p=i;
    if(i>=m)printf("%d ",list1[head].x);
  }
  printf("\n");
  head=1,tail=1;
  list2[1].x=a[1];
  list2[1].p=1;
  for (int i=2;i<=n;i++) {
    while(head<=tail&&i-list2[head].p>=m)head++;
    while(head<=tail&&list2[tail].x<a[i])tail--;
    list2[++tail].x=a[i],list2[tail].p=i;
    if(i>=m)printf("%d ",list2[head].x);
  }
}
```

Basic WC2 exhibits an incorrect order of operations where the answer is retrieved before updating the tail with the current element, causing the current element to be ignored in every window.

---

**Basic WC2 (1111001111)**

```cpp
#include<bits/stdc++.h>
#define ll long long
#define inf 2139062143
#define MAXN 1001000
using namespace std;
inline int read() {
  int x=0,f=1;
  char ch=getchar();
  while(!isdigit(ch)) {
    if(ch=='-') f=-1;
    ch=getchar();
  }
  while(isdigit(ch)) {
    x=x*10+ch-'0',ch=getchar();
  }
  return x*f;
}
int n,m,q[MAXN][2],hd[2],tl[2],a[MAXN],ans[MAXN][2];
int main() {
  n=read(),m=read();
  hd[0]=hd[1]=1;
  for (int i=1;i<=n;i++) {
    a[i]=read();
      while(hd[0]<=tl[0]\&\&q[hd[0]][0]<=i-m) hd[0]++;
      // Swap the order of yellow and red.
    ans[i][0]=a[q[hd[0]][0]];
    while(hd[0]<=tl[0]&&a[q[tl[0]][0]]>=a[i]) tl[0]--;
    q[++tl[0]][0]=i;
    while(hd[1]<=tl[1]&&q[hd[1]][1]<=i-m) hd[1]++;
      // Swap the order of yellow and red.
    ans[i][1]=a[q[hd[1]][1]];
    while(hd[1]<=tl[1]&&a[q[tl[1]][1]]<=a[i]) tl[1]--;
    q[++tl[1]][1]=i;
  }
  for (int i=m;i<n;i++) printf("%d ",ans[i][0]);
  printf("%d\n",ans[n][0]);
  for (int i=m;i<n;i++) printf("%d ",ans[i][1]);
  printf("%d",ans[n][1]);
}
```

---

Distinct from the queue error in Basic WC1, Basic WC3 allocates insufficient memory for the input array using a size of $10^5$ rather than the required $10^6$.

**Basic WC3 (0000001000)**

```cpp
#include<bits/stdc++.h>
using namespace std;
int n,k;
int tail,front;
struct node {
  int pos,val;
}
q[100000010];
int a[100010]; // 100010 -> 1000010
int main() {
  scanf("%d%d",&n,&k);
  for (int i=1;i<=n;i++) {
    scanf("%d",&a[i]);
  }
  front=1;
  tail=1;
  q[1].val=a[1];
  q[1].pos=1;
  for (int i=2;i<=n;i++) {
    while (tail>=front && q[tail].val>=a[i]) tail--;
    q[++tail].val=a[i];
    q[tail].pos=i;
    while (q[tail].pos-q[front].pos+1>k) front++;
    if (i>=k) cout<<q[front].val<<" ";
  }
  cout<<endl;
  front=1;
  tail=1;
  q[1].val=a[1];
  q[1].pos=1;
  for (int i=2;i<=n;i++) {
    while (tail>=front && q[tail].val<=a[i]) tail--;
    q[++tail].val=a[i];
    q[tail].pos=i;
    while (q[tail].pos-q[front].pos+1>k) front++;
    if (i>=k) cout<<q[front].val<<" ";
  }
}
```

Basic WC4 contains a subtle logic error in queue maintenance by performing an unnecessary and erroneous comparison with the head element while updating the tail. This additional operation prevents current elements from entering the queue, causing the queue to potentially become empty during the sliding process. In this state, accessing `q1.top()` triggers undefined behavior, retrieving residual garbage data from the underlying memory address.

---

**Basic WC4 (0001001011)**

```cpp
#include <bits/stdc++.h>
using namespace std;
struct node {
  int x,bh;
  friend bool operator < (node x,node y) {return x.x>y.x;}
} a[1000001];
struct node1 {
  int x,bh;
  friend bool operator < (node1 x,node1 y) {return x.x<y.x;}
} a2[1000001];
priority_queue<node> q1;
priority_queue<node1> q2;
int n,k;
inline int read() {...}
inline void write(int x) {...}
int main() {
  int i;
  n=read();
  k=read();
  int tail=2;
  int head=k+1;
    for (i=1;i<=n;i++) a[i].x=a2[i].x=read(),a[i].bh=a2[i].bh=
        ↪ i;
  for (i=1;i<=k;i++) q1.push(a[i]),q2.push(a2[i]);
  write(q1.top().x);
  printf(" ");
  for (;head<=n;tail++,head++) {
    while(q1.top().bh<tail && !q1.empty()) q1.pop();
    // remove
    if(q1.top().x>=a[head].x) q1.push(a[head]);
    write(q1.top().x);
    printf(" ");
  }
    printf("\n");
  tail=2;
  head=k+1;
  write(q2.top().x);
  printf(" ");
  for (;head<=n;tail++,head++) {
    while(q2.top().bh<tail && !q2.empty()) q2.pop();
    // remove
    if(q2.top().x<=a[head].x) q2.push(a2[head]);
    write(q2.top().x);
    printf(" ");
  }
}
```

---

Basic WC5 represents a scope error where the head and tail pointers of the queue are incorrectly re-initialized inside the loop.

**Basic WC5 (0001000010)**

```c
#include <stdio.h>
#include <stdlib.h>
#define Z 1000001
int main() {
  int i,le=0,ri=1;
  int m,n;
  int *da=(int*)malloc(sizeof(int)*Z);
  int *max=(int*)malloc(sizeof(int)*Z);
  int *min=(int*)malloc(sizeof(int)*Z);
  int *id=(int*)malloc(sizeof(int)*Z);
  scanf("%d",&m);
  scanf("%d",&n);
  for (i=1;i<=m;i++) {
    scanf("%d",&da[i]);
  }
  for (i=1;i<=m;i++) {
    while(le<=ri&&da[i]<min[ri]) {
      ri--;
    }
    ri++;
    min[ri]=da[i];
    id[ri]=i;
    if(id[le]+n<=i) {
      le++;
    }
    if(i>=n) {
      printf("%d ",min[le]);
    }
  }
  printf("\n");
  for (i=1;i<=m;i++) {
    le=0;  // Move outside the loop
    ri=1;
    while(le<=ri&&da[i]>max[ri]) {
      ri--;
    }
    ri++;
    max[ri]=da[i];
    id[ri]=i;
    if(id[le]+n<=i) {
      le++;
    }
    if(i>=n) {
      printf("%d ",max[le]);
    }
  }
  return 0;
}
```

Basic WC6 attempts a Sparse Table (ST) optimization but fails due to an implementation error where the allocated table size is too small for the problem constraints.

**Basic WC6 (0000010101)**

```cpp
#include<bits/stdc++.h>
using namespace std;
const int N = 1e6;
int st[N][18],a[N],p[N]; // 18 -> 20
int maxx[N],minn[N];
int n,k,le,ri;
void init() {
  for (int j = 1; j< 18; j++)
     for (int i = 1; i<=n&& i + ( 1 << j) - 1<=n; ++i)
        st[i][j] = max(st[i][j - 1],st[i + (1<<j - 1)][j -
           ↪ 1]);
}
void init1() {
  for (int j = 1; j< 18; j++)
     for (int i = 1; i<=n&& i + ( 1 << j) - 1<=n; ++i)
        st[i][j] = min(st[i][j - 1],st[i + (1<<j - 1)][j -
           ↪ 1]);
}
int rmq(int l,int r) {
  int d = r - l + 1;
  return max(st[l][p[d]],st[r - (1<<p[d]) + 1][p[d]]);
}
int rmq1(int l,int r) {
  int d = r - l + 1;
  return min(st[l][p[d]],st[r - (1<<p[d]) + 1][p[d]]);
}
int main() {
  scanf("%d%d",&n,&k);
  for (int i = 1; i<=n; i++) {
    scanf("%d",&a[i]);
    st[i][0] = a[i];
  }
  init1();
  for (int i = 1; i<=n; i++) {
    p[i] = p[i-1];
    if(i == 1<<p[i] + 1)
             ++p[i];
  }
  for (int i = 1; i<=n - k+ 1; i++)
        minn[i] = rmq1(i,i + k -1);
  for (int i = 1; i<=n - k + 1; i++)
        cout<<minn[i]<<" ";
  printf("\n");
```

Basic WC7 attempts to fix the boundary error seen in Basic WC6 by incrementing the Sparse Table size by 1, yet it remains insufficient for the maximum constraint.

**Basic WC7 (0000000100)**

```cpp
#include<bits/stdc++.h>
using namespace std;
const int N = 1e6;
int st[N][19],a[N],p[N]; // 19 -> 20
int maxx[N],minn[N];
int n,k,le,ri;
void init() {
  for (int j = 1; j< 19; j++)
     for (int i = 1; i<=n&& i + ( 1 << j) - 1<=n; ++i)
        st[i][j] = max(st[i][j - 1],st[i + (1<<j - 1)][j -
            ↪ 1]);
}
void init1() {
  for (int j = 1; j< 19; j++)
     for (int i = 1; i<=n&& i + ( 1 << j) - 1<=n; ++i)
        st[i][j] = min(st[i][j - 1],st[i + (1<<j - 1)][j -
            ↪ 1]);
}
int rmq(int l,int r) {
  int d = r - l + 1;
  return max(st[l][p[d]],st[r - (1<<p[d]) + 1][p[d]]);
}
int rmq1(int l,int r) {
  int d = r - l + 1;
  return min(st[l][p[d]],st[r - (1<<p[d]) + 1][p[d]]);
}
int main() {
  scanf("%d%d",&n,&k);
  for (int i = 1; i<=n; i++) {
    scanf("%d",&a[i]);
    st[i][0] = a[i];
  }
  init1();
  for (int i = 1; i<=n; i++) {
    p[i] = p[i-1];
    if(i == 1<<p[i] + 1)
           ++p[i];
  }
  for (int i = 1; i<=n - k+ 1; i++)
       minn[i] = rmq1(i,i + k -1);
  for (int i = 1; i<=n - k + 1; i++)
       cout<<minn[i]<<" ";
  printf("\n");
  init();
  for (int i = 1; i<=n; i++)
       st[i][0] = a[i];
  for (int i = 1; i<=n - k + 1; i++)
  maxx[i] = rmq(i,i + k - 1);
  for (int i = 1; i<=n - k + 1; i++)
       cout<<maxx[i]<<" ";
  return 0;
}
```

Basic WC8 incorrectly updates the head pointer instead of the tail pointer during the first element's insertion. Additionally, it omits the insertion of the first element when initializing the second queue.

**Basic WC8 (1100000000)**

```cpp
#include<bits/stdc++.h>
using namespace std;
const int N=1e6+3;
int n,k;
int a[N];
int h=0,t=-1;
int q[N];
int main() {
  cin>>n>>k;
  for (int i=1;i<=n;i++) {
    cin>>a[i];
  }
  q[++h]=1;  // q[++t]=1
  for (int i=2;i<=n;i++) {
    while(i-k+1>q[h]&&h<=t)h++;
    while(h<=t&&a[i]<=a[q[t]])--t;
    q[++t]=i;
    if(i>=k)cout<<a[q[h]]<<" ";
  }
  cout<<endl;
  h=0,t=-1;
    // add q[++t]=1
  for (int i=2;i<=n;i++) {
    while(i-k+1>q[h]&&h<=t)h++;
    while(h<=t&&a[i]>=a[q[t]])--t;
    q[++t]=i;
    if(i>=k)cout<<a[q[h]]<<" ";
  }
  return 0;
}
```

The eight Basic WCs effectively map to the specific requirements of data structures and algorithms inherent to this problem. These error patterns include resource allocation for different variables, as well as the position, order, and conditions for queue initialization and maintenance.

On one hand, Basic WCs cover boundary constraints across different variables and granularities. For instance, Basic WC3 represents a resource error in the *input array* while Basic WC1, Basic WC6, and Basic WC7 target the queue. Specifically, the internal hierarchy among Basic WC1, Basic WC6, and Basic WC7 introduces a tiered validation mechanism. A less advanced test case generator, which could produce medium-scale inputs but struggles with maximum constraints, can still identify Basic WC1 and receive partial credit. This effectively avoids the "all-or-nothing" scoring trap, ensuring that the benchmark gives non-zero scores to generators that possess intermediate capabilities. Conversely, only top-tier generators that hit the absolute maximum boundary can exclude all these Basic WCs to achieve a perfect score.

On the other hand, the basis preserves logic specificities. Basic WC5 incorrectly re-initializes queue pointers inside the loop, causing the state of the sliding window to be lost at every iteration. To expose this, the test case generator must produce inputs where the window's extremum is determined by a historical element rather than the current one, verifying the persistence of the queue. Basic WC8 exhibits dual failures, specifically on the first element's insertion and the second queue's initialization. This forces the generator to produce edge cases where the first element is the strict maximum or minimum for the initial windows. By retaining these basic error patterns, TC-Bench

ensures that the evaluation reflects a model's ability to cover the entire spectrum of the solution space.

## C.2 EXCLUDED WRONG CODES

Following the analysis of the Basic WCs, we proceed to examine the Excluded WCs to verify whether their error patterns are effectively encapsulated by the basis. We select Excluded WC1 as a representative example, whose failure signature is reconstructed by the combination of Basic WC8 and Basic WC4. Excluded WC1 exhibits a classic Off-by-one boundary error. During the sliding process, the code fails to timely pop the element exiting the window, causing the queue to retain invalid, expired data throughout both the initialization and maintenance phases. Crucially, this composite behavior is spanned by the basis. Basic WC8 precisely mirrors the initialization failure, as it retains stale data from the first queue due to a missing head pointer update. Meanwhile, Basic WC4 captures the maintenance failure, where additional comparison causes the queue to become empty. In this state, accessing the queue retrieves residual garbage data from memory. Together, these underlying mechanisms fully cover the error pattern of Excluded WC1.

$$
\begin{array}{r}
\texttt{Basic WC8:1 1 0 0 0 0 0 0 0 0} \\
\texttt{Basic WC4:0 0 0 1 0 0 1 0 1 1} \\
\hline
\texttt{Excluded WC1:1 1 0 1 0 0 1 0 1 1}
\end{array}
$$

**Excluded WC1 (1101001011)**

```cpp
#include<bits/stdc++.h>
using namespace std;
const int N=1000005;
int a,b;
int g[N],num[N],q[N],f1[N],f2[N];
int main() {
  scanf("%d%d",&a,&b);
  for (int i=1;i<=a;i++) {
    scanf("%d",&g[i]);
  }
  int head=1,tail=1;
  for (int i=1;i<=a;i++) {
    while(num[head]<i-b &&head<=tail) // i-b+1
        head++;
    while(g[i]<=q[tail]&&head<=tail) tail--;
    num[++tail]=i;
    q[tail]=g[i];
    f1[i]=q[head];
  }
  head=1,tail=0;
  for (int i=1;i<=a;i++) {
    while(num[head]<i-b+1&&head<=tail) head++;
    while(g[i]>=q[tail]&&head<=tail) tail--;
    num[++tail]=i;
    q[tail]=g[i];
    f2[i]=q[head];
  }
  for (int i=b;i<=a;i++) cout<<f1[i]<<" ";
  cout<<endl;
  for (int i=b;i<=a;i++) cout<<f2[i]<<" ";
  cout<<endl;
  return 0;
}
```

Similarly, we analyze Excluded WC2, whose failure signature corresponds to the combination of Basic WC8 and Basic WC5. Excluded WC2 contains a boundary error in the monotonic queue maintenance. By using the fixed condition $t_¿=1$ instead of the dynamic $h_¡=t$, the tail pointer can incorrectly decrement past the head pointer, violating the valid window scope and accessing invalid historical data. This error pattern is also effectively spanned by the basis. Basic WC8 captures the initialization failure, where the pointers fail to correctly establish the queue's start (updating head instead of tail), reflecting the error's mishandling of the absolute beginning. Basic WC5 captures the scope maintenance failure, where the queue's dynamic state is ignored (resetting pointers inside the loop), mirroring how Excluded WC2 ignores the dynamic head boundary and corrupts the persistent state.

```
    Basic WC5 : 0 0 0 1 0 0 0 0 1 0
    Basic WC8 : 1 1 0 0 0 0 0 0 0 0
Excluded WC2 : 1 1 0 1 0 0 0 0 1 0
```

**Excluded WC2 (1101000010)**

```cpp
#include<iostream>
#include<cstdio>
#include<cstring>
using namespace std;
const int N=1e6+5;
int n,k,a[N],q[N],p[N];
int main() {
  scanf("%d%d",&n,&k);
  for (int i=1;i<=n;i++) scanf("%d",&a[i]);
  int h=1,t=0;
  for (int i=1;i<=n;i++) {
    while(q[t]>a[i]&& t>=1 ) t--;// t>=1 -> h<=t
    q[++t]=a[i],p[t]=i;
    while(p[h]<i-k+1&&h<=t) h++;
    if(i>=k) printf("%d ",q[h]);
  }
  memset(q,0,sizeof(q));
  memset(p,0,sizeof(p));
  cout<<endl;
  h=1,t=0;
  for (int i=1;i<=n;i++) {
    while(q[t]<a[i]&& t>=1 ) t--;// t>=1 -> h<=t
    q[++t]=a[i],p[t]=i;
    while(p[h]<i-k+1&&h<=t) h++;
    if(i>=k) printf("%d ",q[h]);
  }
  return 0;
}
```

## C.3 REPEATED WRONG CODES

Finally, we verified whether identical failure signatures indeed correspond to semantically equivalent error patterns. We selected a cluster of Repeated WCs sharing the binary signature `00010001110`.

We specifically examine the representative example, Repeated WC1, shown below. This code exhibits a logic flaw during the queue maintenance phase: when inserting the current integer, the code compares it against the queue head rather than the queue tail. In a monotonic queue, the tail elements must be compared and popped to maintain monotonicity. Comparing against the head (which typically holds the window's extremum) creates an irrelevant condition. Consequently, elements that should have been removed remain in the queue, corrupting the window's state.

We thoroughly inspected other WCs within this same signature cluster. While they exhibit syntactic variations in implementation, we confirm that they all share the exact same root cause: the failure to correctly remove invalid elements due to flawed comparison logic. This confirms that our signature-based grouping effectively captures semantically similar faults. Due to space constraints, only key segments of these Repeated WCs are presented below.

---

**Repeated WC1 (0001001110)**

```cpp
#include<bits/stdc++.h>
#define maxn 1000010
using namespace std;
int pos[maxn],que[maxn];
int n,k;
int a[maxn];
int fminn[maxn],fmaxx[maxn];
void dpmin() {
  int h = 1, t = 0;
  for (int i = 1; i <= n; i ++) {
    while (pos[h] < i - k + 1 && h <= t) ++ h;
    while (que[t] > a[i] && h <= t) -- t;
    que[++ t] = a[i], pos[t] = i;
    fminn[i] = que[h];
  }
}
void dpmax() {
  int h = 1, t =0;
  for (int i = 1; i <= n; i++) {
    while (pos[h] < i - k + 1 && h <= t) ++ h;
    while ( que[h] < a[i] && h <= t) -- t;// q[h] -> q[t]
    que[++ t] = a[i] , pos[t] = i;
    fmaxx[i] = que[h];
  }
}
int main() {
  scanf("%d%d",&n,&k);
  for (int i = 1; i <= n; i ++) scanf("%d",&a[i]);
  dpmin();
  dpmax();
  for (int i = k; i <= n; i ++) printf("%d ",fminn[i]);
  printf("\n");
  for (int i = k; i <= n; i ++) printf("%d ",fmaxx[i]);
  printf("\n");
  return 0;
}
```

**Repeated WC 2**

```cpp
// Boundary offset error
head = 1; tail = 0;
for (int i = 1; i <= n; ++i) {
    while (head <= tail && i-k-1) head++;// Should be: i-k+1
    while (head <= tail && a[q[tail]] <= a[i]) tail--;
    q[++tail] = i;
    if (i >= k) cout << a[q[head]] << " ";
}
```

**Repeated WC 3**

```cpp
// Incorrectly checks queue tail (r) for expiration
for (int i=k;i<=n;i++) {
  while(l<=r&&a[maxn[r]]<a[i]) r--;
  r++;
  maxn[r]=i;
  while(l<=r && maxn[r] < i-k+1) l++; // Should be: maxn[l]
  cout<<a[maxn[l]]<<" ";
}
```

**Repeated WC 4**

```cpp
// Incorrectly accesses value array 'a' instead of index
    ↪ array 'b'
for (int i=1;i<=n;i++) {
  if(hh<=tt&& a[b[hh]] <=i-k) { // Should be: b[hh]
    hh++;
  }
  while(hh<=tt&&a[b[tt]]<=a[i]) {
    tt--;
  }
}
```

**Repeated WC 5**

```cpp
// Incorrectly compares with queue front while updating tail
for (int i = 1; i <= n; i++) {
  while (!q.empty() && a[ q.front() ] < a[i]) { // Should be:
      ↪ q.back()
    q.pop_back();
  }
  q.push_back(i);
    if (i - q.front() >= m) {
     q.pop_front();
    }
    if (i >= m) cout << a[q.front()] << " ";
}
```

## C.4 DIAGNOSING REALISTIC SCENARIOS

To further verify the diagnostic value of our benchmark in a realistic setting, we conducted an evaluation using the SOTA combination: Claude-4-Thinking with the LCB. We generated 40 test cases. The results showed that the generated test suite successfully excluded 6 out of the 8 Basic WCs but failed to expose Basic WC6 and Basic WC7. They are resource allocation errors requiring queue capacities of approximately $3 \times 10^5$ and $1 \times 10^6$, respectively. Triggering these specific faults requires forcing the monotonic queue to fill up to these limits. Mathematically, this demands a worst-case scenario where both the window size $K$ and the array length $N$ approach $10^6$, and crucially, the input array must follow a specific monotonic pattern (e.g., strictly increasing or decreasing) to

ensure enough elements are pushed into the queue. We manually inspected all 40 generated test cases and confirmed that while the model generated large random arrays, it failed to construct this specific, structurally extreme boundary case. This demonstrates that TC-Bench effectively points out a specific weakness in current SOTA generation methods. Ultimately, this confirms that TC-Bench significantly streamlines diagnostic analysis: by narrowing the analytical scope from the entire raw dataset to a compact set of Basic WCs, it enables researchers to pinpoint model weaknesses through just a few representative examples rather than sifting through massive redundancy.

This case study provides strong empirical evidence for the practical effectiveness of our method. First, the retained Basic WCs are confirmed to be different error patterns. Second, the analysis of Excluded WCs demonstrates that redundant codes are essentially composite errors. Third, the inspection of Repeated WCs confirms that identical failure signatures reliably map to semantically equivalent root causes, validating our signature-based grouping strategy. Fourth, the real-world evaluation highlights the benchmark's discriminative power. Collectively, these results affirm that TC-Bench successfully constructs a compact, rigorous, and representative error space, capable of delivering fine-grained and high-sensitivity evaluations for test case generation.

