# OpenReview forum: "How Many Code and Test Cases Are Enough? Evaluating Test Cases Generation from a Binary-Matrix Perspective"
_ICLR.cc/2026/Conference — ICLR 2026 Poster_

### Official Review · Reviewer_D88H · 2025-10-26

**Soundness:** 3
**Presentation:** 3
**Contribution:** 3
**Rating:** 6
**Confidence:** 3

**Summary:**

This work addresses the challenge of evaluating automatically generated test cases for code using LLMs. The authors propose a new framework that models the relationship between code and test cases as a binary matrix, where the matrix rank reveals the minimal number of distinct wrong codes and test cases needed for evaluation. They introduce WrongSelect, an efficient algorithm for selecting a maximally diverse and representative set of wrong codes to form a compact benchmark, TC-Bench. Built from millions of competitive programming submissions, TC-Bench offers an efficient, diverse, and inflation-resistant benchmark for assessing test-case generation. Evaluation across 13 leading LLMs show that even the best current methods achieve only around 60% fault detection, highlighting significant room for improvement in automated test generation and evaluation.

**Strengths:**

- Formulating problem as code-test matrix
- Number of evaluated LLMs
- Approximation algorithm
- Benchmark construction

**Weaknesses:**

- Source of data from programming contests
- Theoritically ok, but what about practicality?
- Not studying other similarity metrics

**Questions:**

- How practical is this technique in practice/industry?
- Do you think your technique generalizes on other datasets, when your evaluation is only using programming contests?
- Do you think other similarity metrics would fit better than Jaccard?

---

> ### Author Response · Authors · 2025-11-22
>
> We sincerely thank the reviewer for recognizing the soundness of our matrix formulation and the scale of our evaluation!
>
> ---
> ### **Generalizability of Our Methodology**
>
> We targeted **programming contests** because they require LLMs to comprehend problem statements, abstract mathematical models, design data structures, and implement algorithms that meet complexity constraints. Therefore, they **are regarded as a crucial indicator of LLM intelligence and a primary focus of current Code-RL research**. However, this does not imply our method is limited to competitive programming. Since the functional correctness of code is fundamentally verified via test cases, our framework is highly generalizable to software engineering contexts. **In real-world development, historical buggy commits correspond to "Wrong Codes," while accumulated unit tests serve as "Golden Tests"**. Since employing the full dataset for regression testing is often inefficient due to redundancy, our WrongSelect algorithm is directly applicable. By constructing a Commit-Test matrix, it filters out the most representative buggy commits as an "error basis" to create a highly efficient test suite. Furthermore, **any domain utilizing checklist-based evaluations[1] can also adapt our matrix-based selection to optimize their test sets**.
>
> ---
> ### **An Example to illustrate practical effectiveness**
>
> To demonstrate the practical effectiveness of our method beyond theoretical rank preservation, we conducted a comprehensive case study on the classic problem ``Sliding Window'' in Appendix C, which includes the original code, fixes, and detailed analysis. We meticulously examined the retained Basic WCs to confirm their error patterns are mutually distinct, effectively spanning the error space. We further analyzed the Excluded WCs to verify they are indeed composite redundancies derived from the basis. Finally, we confirmed that Repeated WCs sharing the same failure signature stem from identical root causes.
> To contextualize these findings, we first briefly describe the problem logic. Given an input array of length $N (\le 10^6)$ and a window size $K(\le 10^6)$, the task is to find the maximum and minimum values in the window as it slides. The correct solution employs a Monotonic Queue. For example, to find the maximum, we maintain a monotonically decreasing queue. As we iterate through the input array, we first pop the front if the index is out of the window. Then, we pop elements from the back if they are smaller than the current element, since these "smaller and older" elements can never be the maximum. Finally, we push the current element, and the front of the queue represents the maximum in current window.
> Initially, the dataset contained 96 Wrong Codes. After applying WrongSelect, only 8 Basic WCs were retained. Their failure signatures is
> $$
> \mathcal{I}^{*} =
> \begin{bmatrix}
> 0 & 0 & 0 & 0 & 0 & 0 & 0 & 1 & 1 & 0 \\\\
> 1 & 1 & 1 & 1 & 0 & 0 & 1 & 1 & 1 & 1 \\\\
> 0 & 0 & 0 & 0 & 0 & 0 & 1 & 0 & 0 & 0 \\\\
> 0 & 0 & 0 & 1 & 0 & 0 & 1 & 0 & 1 & 1 \\\\
> 0 & 0 & 0 & 1 & 0 & 0 & 0 & 0 & 1 & 0 \\\\
> 0 & 0 & 0 & 0 & 0 & 1 & 0 & 1 & 0 & 1 \\\\
> 0 & 0 & 0 & 0 & 0 & 0 & 0 & 1 & 0 & 0 \\\\
> 1 & 1 & 0 & 0 & 0 & 0 & 0 & 0 & 0 & 0
> \end{bmatrix}
> $$.
> We carefully analyze their original codes and characterize their error patterns.
> - Basic WC1 fails due to insufficient memory allocation for the queue array.
> - Basic WC2 exhibits an incorrect order of operations where the answer is retrieved before updating the tail with the current element, causing the current element to be ignored in every window.
> - Basic WC3 allocates insufficient memory for the input array.
> - Basic WC4 contains a subtle logic error in queue maintenance by performing an erroneous comparison with the head element while updating the tail. This additional operation prevents current elements from entering the queue, causing the queue to potentially become empty during the sliding process. In this state, accessing the element in queue triggers undefined behavior, retrieving residual garbage data from the underlying memory address.
> - Basic WC5 represents a scope error where the head and tail pointers of the queue are incorrectly re- initialized inside the loop.
> - Basic WC6 attempts a sparse table optimization but fails due to an implementation error where the allocated table size is too small for the problem constraints.
> - Basic WC7 attempts to fix the boundary error seen in Basic WC6 by incrementing the sparse table size by 1, yet it remains insufficient for the maximum constraint.
> - Basic WC8 incorrectly updates the head pointer instead of the tail pointer during the first element's insertion. Additionally, it omits the insertion of the first element when initializing the second queue.
>
> **These distinct error patterns form a basis that covers two main categories: resource allocation for different variables, and the management of queues.**

---

> ### Author Response · Authors · 2025-11-22
>
> On one hand, Basic WCs cover boundary constraints across different variables and granularities. WC3 allocates insufficient memory for the Input Array, while WC1, WC6, and WC7 allocate insufficient memory for the Queue. Crucially, WC1, WC6, and WC7 form an internal hierarchy approaching the absolute limit of $10^6$. This structure avoids "all-or-nothing" scoring: a generator producing medium-scale inputs can identify WC6 and receive partial credit, whereas only top-tier generators producing inputs at the absolute limit can identify all.
>
> On the other hand, the basis preserves logic specificities. Basic WC5 incorrectly re-initializes queue pointers inside the loop, causing the state of the sliding window to be lost at every iteration. To expose this, the test case generator must produce inputs where the window's extremum is determined by a historical element rather than the current one, verifying the persistence of the queue. Basic WC8 exhibits dual failures, specifically on the first element's insertion and the second queue's initialization. This forces the generator to produce edge cases where the first element is the strict maximum or minimum for the initial windows.
>
> By retaining these basic error patterns, TC-Bench ensures that the evaluation reflects a model's ability to cover the entire spectrum of the solution space.
>
> **We analyzed Excluded WC1 (and Excluded WC2 in Appendix C) to verify that the removed codes are truly redundant.** Excluded WC1 contains an Off-by-one boundary error, failing to pop the element from the queue that just slid out of the window. This causes the queue to retain expired data, leading to both initialization and maintenance errors. Its failure signature ($1 1 0 1 0 0 1 0 1 1$) is exactly the linear combination of Basic WC8 ($1 1 0 0 0 0 0 0 0 0$) and Basic WC4 ($0 0 0 1 0 0 1 0 1 1$). Basic WC8 captures the initialization/boundary failure (retaining invalid initial data), while Basic WC4 captures the maintenance failure (reading garbage data due to restrictive logic). Therefore, any test suite capable of detecting the specific boundary failure in WC8 and the logic failure in WC4 will automatically detect the composite error in Excluded WC1.
>
> $$
> \begin{aligned}
> \texttt{Basic WC8} & : \texttt{1 1 0 0 0 0 0 0 0 0} \\\\
> \texttt{Basic WC4} & : \texttt{0 0 0 1 0 0 1 0 1 1} \\\\
> \hline
> \texttt{Excluded WC1} & : \texttt{1 1 0 1 0 0 1 0 1 1}
> \end{aligned}
> $$
>
> We examined a cluster of WCs sharing the same binary signature (00010001110). The representative code compares the current element against the Queue Head instead of the Queue Tail, breaking monotonicity. While other codes in this cluster exhibit syntactic variations in implementation, they all share this exact root cause: failing to remove invalid elements due to flawed comparison logic. **This confirms that our signature-based grouping accurately captures semantic equivalence**.
>
> **Diagnosing Realistic Scenarios**
>
> To further verify the diagnostic value of our benchmark in a realistic setting, we conducted an evaluation using the SOTA combination: Claude-4-Thinking with the LCB. We generated 40 test cases.
> The results showed that the generated test suite successfully excluded 6 out of the 8 Basic WCs but failed to expose Basic WC6 and Basic WC7.
> They are resource allocation errors requiring queue capacities of approximately $3 \times 10^5$ and $1 \times 10^6$, respectively. Triggering these specific faults requires forcing the monotonic queue to fill up to these limits. Mathematically, this demands a worst-case scenario where both the window size $K$ and the array length $N$ approach $10^6$, and crucially, the input array must follow a specific monotonic pattern (e.g., strictly increasing or decreasing) to ensure enough elements are pushed into the queue.
> We manually inspected all 40 generated test cases and confirmed that while the model generated large random arrays, it failed to construct this specific, structurally extreme boundary case. This demonstrates that **TC-Bench effectively points out a specific weakness in current SOTA generation methods**.
>
> Ultimately, this confirms that TC-Bench significantly streamlines diagnostic analysis: by narrowing the analytical scope from the entire raw dataset to a compact set of Basic WCs, it enables researchers to analyze weaknesses through just a few representative examples rather than sifting through massive redundancy.
>
> This case study affirms that TC-Bench successfully constructs a compact, rigorous, and representative error space, capable of delivering fine-grained and high-sensitivity evaluations for test case generation.

---

> ### Author Response · Authors · 2025-11-22
>
> ## **Rationale for Choosing Jaccard Similarity**
> **We selected Jaccard Similarity after rigorously evaluating and discarding alternatives like Variance, Hamming Distance, and Cosine Similarity.** Our objective is to ensure that each selected wrong code represents an independent error pattern, avoiding overlap. Ideally, we aim to find a basis in the Code-Test binary matrix that is closest to the standard orthogonal basis (Identity Matrix) shown below:
> $$
> \begin{bmatrix}
> 1 & 0 & 0 & 0 \\\\
> 0 & 1 & 0 & 0 \\\\
> 0 & 0 & 1 & 0 \\\\
> 0 & 0 & 0 & 1
> \end{bmatrix}
> $$
> This implies that each wrong code represents a pure, single error pattern with zero intersection. However, since such an ideal basis rarely exists in reality because failure patterns often overlap, **our objective shifts to finding a basis among all candidates that maximizes orthogonality**.
>
> **Attempt 1: Minimizing Variance** We first attempted to minimize the variance of column sums, motivated by the idea that if error codes are distributed "uniformly," each test case should be covered roughly equally. This would theoretically avoid "hack" problems where generating a single test case captures a disproportionate amount of the score, such as $t_3$ below, which alone secures 75% of the score:
> $$
> \begin{bmatrix} 1 & 0 & 1 & 0 \\\\ 0 & 1 & 1 & 0 \\\\ 0 & 0 & 1 & 0 \\\\ 0 & 0 & 0 & 1 \end{bmatrix}
> $$
> However, **variance only considers uniformity and ignores the density of failures**. For instance, the matrix below has zero variance because all column sums are 3, but it represents highly redundant error patterns, which contradicts our goal of minimizing overlap:
> $$
> \begin{bmatrix} 1 & 1 & 1 & 0 \\\\ 1 & 1 & 0 & 1 \\\\  1 & 0 & 1 & 1 \\\\  0 & 1 & 1 & 1 \end{bmatrix}
> $$
>
> **Attempt 2: Maximizing Hamming Distance** Instead of relying on column statistics, we shifted our focus to measuring the direct pairwise similarity between vectors. Consequently, we considered maximizing the average Hamming distance, which counts the number of positions where corresponding values differ. However, error patterns are only defined by '1's, which represent failures. We aim to avoid 1-1 overlaps, but 0-0 overlaps where both codes pass the same test do not imply that they share the same error pattern. Ideally, independent vectors like $[0,0,1,0]$ and $[0,1,0,0]$ have a Hamming distance of only 2, which is numerically identical to highly overlapping vectors like $[1,0,1,1]$ and $[1,1,0,1]$. Since this metric **cannot distinguish between "independent" and "redundant" pairs in our context**, it is unsuitable.
>
> **Attempt 3: Minimizing Cosine Similarity** To address the 0-0 trap, we tested Cosine Similarity, as the dot product focuses only on 1-1 overlaps. However, **the normalization term, which calculates the geometric mean of magnitudes, introduces bias when comparing vectors with different numbers of '1's**. This is particularly problematic in containment scenarios where the failures of one vector are a subset of another. Consider a basis vector $A$ and two candidates $B$ and $C$:
>
> $$
> \begin{aligned}
> A&=[1,1,1,1,1,0,0,0], \\\\
> B&=[1,1,0,0,0,1,1,1] \(\text{Overlap with A}\), \\\\
> C&=[0,0,0,0,1,0,0,0]  \(\text{Subset of A}\),
> \end{aligned}
> $$
>
> Calculating the similarities:
> $$Cos(A,B) = \frac{2}{\sqrt{5}\sqrt{5}} = 0.4, \quad Jaccard(A,B) = \frac{2}{8} = 0.25$$
> $$Cos(A,C) = \frac{1}{\sqrt{5}\sqrt{1}} \approx 0.4472, \quad Jaccard(A,C) = \frac{1}{5} = 0.2$$
>
> In this scenario, we clearly prefer the independent vector $C$  as it has lower overlap, representing a sparser and more independent error. However, Cosine Similarity $0.4472 > 0.4$ incorrectly suggests $C$ is more similar to $A$ than $B$ is, potentially causing the algorithm to miss $C$. In contrast, Jaccard Similarity $0.2 < 0.25$ correctly identifies $C$ as less similar, effectively handling inclusion relationships. Thus, we selected Jaccard as the optimal metric.
>
> ---
>
> We thank the reviewer again for these valuable and constructive comments, which have helped us significantly clarify the practical applicability of our framework and the theoretical justification for our metric selection!
>
> [1] Gunjal, Anisha, et al. "Rubrics as Rewards: Reinforcement Learning Beyond Verifiable Domains." NeurIPS 2025 Workshop on Efficient Reasoning.

---

> > ### Comment · Reviewer_D88H · 2025-11-23
> >
> > Thank you for addressing my concerns.

---

> > > ### Author Response · Authors · 2025-11-24
> > >
> > > We sincerely appreciate your response. We are encouraged to hear that our clarifications have successfully addressed your concerns.
> > >
> > > We want to briefly highlight that, per your constructive suggestions, we have discussed the generalization of our framework beyond competitive programming, clarified its practicality, and justified the choice of similarity metrics.
> > >
> > > We believe the manuscript is now significantly stronger thanks to your suggestions. We would be grateful if you could consider these updates in your final assessment of our work.

---

### Official Review · Reviewer_tPeL · 2025-10-29

**Soundness:** 4
**Presentation:** 3
**Contribution:** 3
**Rating:** 8
**Confidence:** 4

**Summary:**

This paper addresses how to rigorously evaluate LLM-generated test cases by reframing benchmark construction as a binary matrix problem over wrong codes (rows) and golden tests (columns). The central idea is to select a compact yet representative set of wrong codes that preserves the matrix’s diagnostic capacity while avoiding redundancy and score inflation. Concretely, the authors require the kept rows to form a row basis whose size equals the matrix rank (capturing all independent error modes), and among all such bases, they choose one that maximizes diversity by minimizing average pairwise Jaccard similarity of failure signatures.

They propose WrongSelect, an efficient approximation combining principled pre-filtering and random-restart local search. Pre-filtering drops problems with any all-ones test column (hack-prone) and removes trivial wrong codes that fail ≥80% of tests; then the local search swaps rows in/out while maintaining rank to minimize the diversity objective.

**Strengths:**

1. The paper presents a fresh and elegant formulation that connects test selection to linear algebra.

Modeling wrong-code failure signatures as rows in a binary matrix and selecting a rank-preserving, diversity-maximizing row basis is a principled way to eliminate redundancy while retaining all independent error modes. This perspective clarifies what “good coverage” means and gives a theoretical upper bound on minimal tests.

2. The empirical results are strong.

The proposed selection method yields a compact, diverse benchmark that meaningfully stresses current test generators, and the reported exclusion rates show clear, consistent improvements over baselines. The ablations and end-to-end evaluations convincingly demonstrate both effectiveness and practicality at scale.

**Weaknesses:**

1. The approach assumes access to a sufficiently rich pool of golden tests (public + private) to build informative failure profiles.

In many real-world settings, curating or synthesizing high-quality golden tests is costly and time-consuming. This reliance reduces the method’s portability and makes it difficult for third parties to reproduce or extend the benchmark to new domains or problem sets without significant investment.

2. While the formulation is theoretically sound (rank preservation plus diversity), the paper provides limited qualitative analysis to illustrate practical effectiveness.

Concrete case studies—e.g., representative wrong-code examples retained vs. removed, error modes uncovered by the selected basis, or developer-facing insights derived from the reduced matrix—would strengthen the claim that the method improves understanding and diagnosis beyond raw metrics.

**Questions:**

could you provide an example for the effectiveness of your approach in a code example?

---

> ### Author Response · Authors · 2025-11-22
>
> We sincerely thank the reviewer for both highly constructive and inspiring comments! We are deeply encouraged by your appreciation of our novel formulation connecting test selection to linear algebra. Below, we address the reliance on Golden Tests and provide the detailed case study you requested.
>
> ---
>
> ### **Generalizability of Our Methodology**
> We acknowledge that in emerging domains, Golden Test Cases can be scarce, and evaluation often relies on white-box metrics like line/branch coverage[1]. However, we highlight a fundamental fact: **the functional correctness of code can only be verified through sufficient GTs**. If a domain lacks these basic correctness standards, the primary challenge lies in establishing ground truth rather than optimizing evaluation. Our method is strategically positioned for the subsequent stage: efficient cleaning, deduplication, and benchmark construction once raw data is accumulated.
>
> Our methodology possesses strong generalizability and maps directly to software engineering scenarios. In large-scale projects (e.g., popular GitHub repos), **historical buggy commits correspond to "Wrong Codes," while accumulated unit tests serve as "Golden Tests."** Since employing the full dataset for testing is often inefficient due to redundancy, our WrongSelect algorithm can be applied here: by constructing a Commit-Test matrix, it filters out the most representative buggy commits as an "error basis," creating a highly efficient test suite.
>
> **Given that acquiring GTs is resource-intensive**, it is critical to prevent them from being underutilized or obscured by massive redundancy during evaluation. **The core value of TC-Bench lies in maximizing the ROI (Return on Investment) of these high-cost data**. By mathematically extracting core error patterns, we enable future researchers to reuse this high-quality diagnostic information at minimal cost.
>
> ---
>
> ### **An Example to illustrate practical effectiveness**
>
> To demonstrate the practical effectiveness of our method beyond theoretical rank preservation, we conducted a comprehensive case study on the classic problem ``Sliding Window'' in Appendix C, which includes the original code, fixes, and detailed analysis. We meticulously examined the retained Basic WCs to confirm their error patterns are mutually distinct, effectively spanning the error space. We further analyzed the Excluded WCs to verify they are indeed composite redundancies derived from the basis. Finally, we confirmed that Repeated WCs sharing the same failure signature stem from identical root causes.
>
> To contextualize these findings, we first briefly describe the problem logic. Given an input array of length $N (\le 10^6)$ and a window size $K(\le 10^6)$, the task is to find the maximum and minimum values in the window as it slides. The correct solution employs a Monotonic Queue. For example, to find the maximum, we maintain a monotonically decreasing queue. As we iterate through the input array, we first pop the front if the index is out of the window. Then, we pop elements from the back if they are smaller than the current element, since these "smaller and older" elements can never be the maximum. Finally, we push the current element, and the front of the queue represents the maximum in current window.
> Initially, the dataset contained 96 Wrong Codes. After applying WrongSelect, only 8 Basic WCs were retained. Their failure signatures is
> $$
> \mathcal{I}^{*} =
> \begin{bmatrix}
> 0 & 0 & 0 & 0 & 0 & 0 & 0 & 1 & 1 & 0 \\\\
> 1 & 1 & 1 & 1 & 0 & 0 & 1 & 1 & 1 & 1 \\\\
> 0 & 0 & 0 & 0 & 0 & 0 & 1 & 0 & 0 & 0 \\\\
> 0 & 0 & 0 & 1 & 0 & 0 & 1 & 0 & 1 & 1 \\\\
> 0 & 0 & 0 & 1 & 0 & 0 & 0 & 0 & 1 & 0 \\\\
> 0 & 0 & 0 & 0 & 0 & 1 & 0 & 1 & 0 & 1 \\\\
> 0 & 0 & 0 & 0 & 0 & 0 & 0 & 1 & 0 & 0 \\\\
> 1 & 1 & 0 & 0 & 0 & 0 & 0 & 0 & 0 & 0
> \end{bmatrix}
> $$.
>
> We carefully analyze their original codes and characterize their error patterns.

---

> ### Author Response · Authors · 2025-11-22
>
> - Basic WC1 fails due to insufficient memory allocation for the queue array.
> - Basic WC2 exhibits an incorrect order of operations where the answer is retrieved before updating the tail with the current element, causing the current element to be ignored in every window.
> - Basic WC3 allocates insufficient memory for the input array.
> - Basic WC4 contains a subtle logic error in queue maintenance by performing an erroneous comparison with the head element while updating the tail. This additional operation prevents current elements from entering the queue, causing the queue to potentially become empty during the sliding process. In this state, accessing the element in the queue triggers undefined behavior, retrieving residual garbage data from the underlying memory address.
> - Basic WC5 represents a scope error where the head and tail pointers of the queue are incorrectly re-initialized inside the loop.
> - Basic WC6 attempts a sparse table optimization but fails due to an implementation error where the allocated table size is too small for the problem constraints.
> - Basic WC7 attempts to fix the boundary error seen in Basic WC6 by incrementing the sparse table size by 1, yet it remains insufficient for the maximum constraint.
> - Basic WC8 incorrectly updates the head pointer instead of the tail pointer during the first element's insertion. Additionally, it omits the insertion of the first element when initializing the second queue.
>
> **These distinct error patterns form a basis that covers two main categories: resource allocation for different variables, and the management of queues.**
>
> On one hand, Basic WCs cover boundary constraints across different variables and granularities. WC3 allocates insufficient memory for the Input Array, while WC1, WC6, and WC7 allocate insufficient memory for the Queue. Crucially, WC1, WC6, and WC7 form an internal hierarchy approaching the absolute limit of $10^6$. This structure avoids "all-or-nothing" scoring: a generator producing medium-scale inputs can identify WC6 and receive partial credit, whereas only top-tier generators producing inputs at the absolute limit can identify all.
>
> On the other hand, the basis preserves logic specificities. Basic WC5 incorrectly re-initializes queue pointers inside the loop, causing the state of the sliding window to be lost at every iteration. To expose this, the test case generator must produce inputs where the window's extremum is determined by a historical element rather than the current one, verifying the persistence of the queue. Basic WC8 exhibits dual failures, specifically on the first element's insertion and the second queue's initialization. This forces the generator to produce edge cases where the first element is the strict maximum or minimum for the initial windows.
>
> By retaining these basic error patterns, TC-Bench ensures that the evaluation reflects a model's ability to cover the entire spectrum of the solution space.
>
> **We analyzed Excluded WC1 (and Excluded WC2 in Appendix C) to verify that the removed codes are truly redundant.** Excluded WC1 contains an Off-by-one boundary error, failing to pop the element from the queue that just slid out of the window. This causes the queue to retain expired data, leading to both initialization and maintenance errors. Its failure signature ($1 1 0 1 0 0 1 0 1 1$) is exactly the linear combination of Basic WC8 ($1 1 0 0 0 0 0 0 0 0$) and Basic WC4 ($0 0 0 1 0 0 1 0 1 1$). Basic WC8 captures the initialization/boundary failure (retaining invalid initial data), while Basic WC4 captures the maintenance failure (reading garbage data due to restrictive logic). Therefore, any test suite capable of detecting the specific boundary failure in WC8 and the logic failure in WC4 will automatically detect the composite error in Excluded WC1.
> $$
> \begin{aligned}
> \texttt{Basic WC8} & : \texttt{1 1 0 0 0 0 0 0 0 0} \\\\
> \texttt{Basic WC4} & : \texttt{0 0 0 1 0 0 1 0 1 1} \\\\
> \hline
> \texttt{Excluded WC1} & : \texttt{1 1 0 1 0 0 1 0 1 1}
> \end{aligned}
> $$
>
> We examined a cluster of WCs sharing the same binary signature (00010001110). The representative code compares the current element against the Queue Head instead of the Queue Tail, breaking monotonicity. While other codes in this cluster exhibit syntactic variations in implementation, they all share this exact root cause: failing to remove invalid elements due to flawed comparison logic. **This confirms that identical failure signatures indeed correspond to semantically equivalent error patterns**.

---

> ### Author Response · Authors · 2025-11-22
>
> **Diagnosing Realistic Scenarios**
>
> To further verify the diagnostic value of our benchmark in a realistic setting, we conducted an evaluation using the SOTA combination: Claude-4-Thinking with the LCB. We generated 40 test cases.
> The results showed that the generated test suite successfully excluded 6 out of the 8 Basic WCs but failed to expose Basic WC6 and Basic WC7.
> They are resource allocation errors requiring queue capacities of approximately $3 \times 10^5$ and $1 \times 10^6$, respectively. Triggering these specific faults requires forcing the monotonic queue to fill up to these limits. Mathematically, this demands a worst-case scenario where both the window size $K$ and the array length $N$ approach $10^6$, and crucially, the input array must follow a specific monotonic pattern (e.g., strictly increasing or decreasing) to ensure enough elements are pushed into the queue.
> We manually inspected all 40 generated test cases and confirmed that while the model generated large random arrays, it failed to construct this specific, structurally extreme boundary case. This demonstrates that **TC-Bench effectively points out a specific weakness in current SOTA generation methods**.
>
> Ultimately, this confirms that TC-Bench significantly streamlines diagnostic analysis: by narrowing the analytical scope from the entire raw dataset to a compact set of Basic WCs, it enables researchers to analyze weaknesses through just a few representative examples rather than sifting through massive redundancy.
>
> This case study affirms that TC-Bench successfully constructs a compact, rigorous, and representative error space, capable of delivering fine-grained and high-sensitivity evaluations for test case generation.
>
> ---
>
> We are truly grateful for your insightful feedback. Your suggestion to include a concrete case study was invaluable, helping us to strengthen the depth of our paper significantly. Your guidance has greatly elevated the final quality of our work!

---

> ### Author Response · Authors · 2025-11-25
> **Kindly Reminder**
>
> Dear Reviewer:
>
> Has our reply resolved your concern? If you have any further questions, please don't hesitate to ask, and we'll be happy to respond. We deeply appreciate your feedback, as it is invaluable for improving the quality of our work.

---

> > ### Author Response · Authors · 2025-11-27
> > **Kindly Reminder**
> >
> > Dear Reviewer:
> >
> > Has our reply resolved your concern? If you have any further questions, please don't hesitate to ask, and we'll be happy to respond. We deeply appreciate your feedback, as it is invaluable for improving the quality of our work.

---

### Official Review · Reviewer_jxDY · 2025-10-30

**Soundness:** 3
**Presentation:** 2
**Contribution:** 4
**Rating:** 6
**Confidence:** 4

**Summary:**

This paper proposes a new benchmark to evaluate LLMs' ability to generate high quality test cases for coding problems. The quality is determined by the efficiency in covering the error space of the problem. To improve quality representation of the error space, the authors form an error matrix, consisting of all the submissions by their pass/fail results on all the test cases. Since the rank of this 2D matrix conveniently represents the "coverage" of submission patterns, they develop a greedy algorithm to maximize the diversity of wrong code submission, which in return reduces to a more compact submission set for efficient evaluation.
The authors then evaluate several test case generation methods on various LLMs on two major metrics – the accuracy of generated test cases (PR) and the degree of successful filtering of wrong code submissions (HR). The evaluation results show that the choice of generation method matters significantly more than the choice of the underlying LLM. At the same time, the relatively underwhelming numbers (best combo leads to 60~% in HR) points to the importance of constructing a high quality set of wrong code – the analysis on the unfiltered version of the dataset reveals inflated scores, which is inaccurate.

**Strengths:**

* A meaningful contribution for a much-needed area, both in terms of efficiency and framework. The method can see practical use cases.
* The proposed method for selecting the minimal set of wrong code submissions is principled and reasonable.
* The outcome dataset effectively pinpoints the need for better TCG methods, citing the underwhelming numbers.

**Weaknesses:**

* Apple-to-apple comparison against the previous work: It would be stronger if you could replicate the other test case evaluation methods on the same problem set and show how TC-Bench more critically measures the TCG quality by LLMs. I'm aware this is done partially by comparing the method against the "All WC" counterpart.
* WrongSelect's robustness: It's uncertain if the greedy algorithm yields a stable set of problems.
* The need of translation: I assume the dataset is entirely sourced from non-English problems. The choice of data sources affect the quality and the number of (valid) wrong code submissions for problems. Have authors explored English data? Reducing the submission count to less than 2% depends on the population of wrong code submissions.

---

Below are NOT the weaknesses of the method, but I think they are important to raise for a higher quality submission. I'm leaving here as I don't see fit in "Questions"
* Presentation 1: Overall there is high usage of acronyms, such as "some WCs labeled as WA under GTs produce RE or TLE when executed on ATs". I got used to it by the end a little bit, but I often had to go back and refresh my glossary many times. I'd highly suggest fixing the inconsistency (e.g., "wrong code" and WC) and reducing overall abbreviations.
* Presentation 2: Plot styles are inconsistent – mostly comic sans but some of them serif (Times?) out of a sudden. It looks like being patched together. I recommend following a single style.
* Appendix B has no content

**Questions:**

* How much variance in PR / HR do you observe by running the optimization multiple times? When you say convergence, do they arrive at the same set of WCs?
* The analysis comparing the method against the "All WCs", are the all WCs before or after pre-filtering?

---

> ### Author Response · Authors · 2025-11-20
>
> We sincerely thank the reviewer for recognizing our work as a meaningful contribution to the area!
>
> ---
>
> ### **Apple-to-apple Comparison against Previous Work**
>
> We selected three representative baselines for a rigorous comparison on the Claude-4-thinking model: TCG [1], which randomly selects 5 wrong codes from those passing at least 60% of test cases. TestCase-Eval [2], which randomly samples 20 wrong codes per problem. TCGBench [3] (All_WC), which uses the full collection of wrong codes. The results are presented in the table below (we also update the Figure in the paper):
>
> | Methods | TCG   | TestCase-Eval | ALL_WC(TCGBench)| TC-Bench(Ours) |
> |---------|-------|---------------|---------------|----------------|
> | CRUX    | 32.86 | 46.96         | 56.74         | 30.45          |
> | PREDSUDO   | 10.26 | 20.7          | 23.91         | 12.6           |
> | ALGO    | 35.16 | 56.24         | 61.69         | 37.07          |
> | LCB     | 47.11 | 81.15         | 86.71         | 54.14          |
> | HT      | 41.06 | 79.04         | 81.93         | 53.28          |
>
> As observed, TestCase-Eval exhibits scores and trends highly similar to ALL_WC across all augment test cases. This indicates that randomly sampling 20 codes, while potentially picking some unique ones, is fundamentally a scaling operation. It fails to exclude redundant wrong codes and thus cannot resolve the issue of score inflation. TCG lowers scores by enforcing a heuristic pass-rate threshold (0.4) and a fixed size limit (5). However, this heuristic is inflexible, leading to insufficient coverage of error patterns.
>
> To further analyze this, we compared the rank distribution of wrong codes retained by each method against the original rank for each problem (Ours) in Figure 4(b) in the paper. To minimize stochasticity, we averaged the rank distribution of wrong codes from 5 random sampling runs. The results show that the rank of the error space varies significantly per problem. While most are below 20, some approach 30. The heuristic uniform sampling of TestCase-Eval results in significant redundancy for low-rank problems. Even for high-rank problems, it cannot guarantee that the sampled codes represent independent error patterns. Conversely, TCG drastically reduces redundancy by sampling only 5 codes. While this lowers the performance scores of current methods, it paradoxically makes future methods prone to score inflation: they would only need to cover a maximum subset of five patterns rather than the complete error space.
>
> In summary, TC-Bench strikes the optimal balance, avoiding both the inflation of coverage-based methods and the under-representation of heuristic random sampling.

---

> ### Author Response · Authors · 2025-11-20
>
> ### **Stability and Robustness of WrongSelect**
>
> We thank the reviewer for this inspiring question, which prompted us to deepen our theoretical analysis. WrongSelect filters wrong codes for each problem independently, so it does not affect the set of problems itself. We repeated WrongSelect 5 times and found that **for every problem, the final set of wrong codes was identical across all runs. Therefore, our convergence results in exactly the same set of wrong codes, and the HR/PR metrics remain consistent**.
>
> We attribute this stability to the data characteristics and our algorithm design. Our objective is to find the basis with the lowest Jaccard Similarity, which is highly sensitive to overlap. After WrongSelect starts, once a redundant vector is replaced by an independent one (which is orthogonal or has a large angle to the existing basis), the objective function value drops sharply. Since the inner loop iterates through all external vectors, **the steep improvement gain provided by independent vectors ensures they are selected early in the process**. Furthermore, **independent vectors are mostly orthogonal to each other. Selecting one does not preclude selecting another, nor does it trap the algorithm in a local optimum. Consequently, the algorithm quickly aggregates these independent vectors in the first few steps.** Replacing any of these independent vectors with another vector would increase overlap and raise the function value, so **the algorithm will not swap them out once collected**. This implies that regardless of the starting basis, these independent vectors will rapidly attract the basis towards them. **Since these independent vectors are definitely in the optimal solution, our algorithm converges quickly**, as shown in Figure 9. **The structure of the solution is inherently determined by the data**: there exists a vast number of wrong codes containing specific error patterns. These codes pass most test cases and only fail on distinct, rare corner cases. Moreover, we employ 1,000 random restarts. Assuming the probability of a greedy search failing to find the global optimum is $P_{fail}$, the probability of failing 1,000 times is ${P_{fail}}^{1000}$. Even if $P_{fail}=0.99$, $0.99^{1000} \approx 0.00004$. This large-scale computation provides probabilistic certainty. **Since the random seed does not determine the solution structure, our greedy algorithm finds the global optimum rather than getting stuck in local optima with similar scores.**
>
> ---
>
> ### **Multilingual Data Source**
>
> We apologize for the misunderstanding that our data source is limited to Chinese. We used Chinese in the illustrations solely to demonstrate the translation step of our data processing pipeline. The dataset includes English problems from international sources like ICPC Regional Contests and Olympiads in Informatics from European countries (line 112-113 in paper). We carefully vetted sources to ensure high-quality, competitive-level data and gathered as many submissions as possible. The "less than 2%" retention rate is a statistical outcome of our algorithm rather than a predefined target. We prioritize diversity and quality, and the algorithm naturally reduced the dataset to 2% after filtering redundancy. We believe the methodology behind TC-Bench is generalizable and look forward to seeing teams with access to even larger proprietary repositories adopt our method to build better benchmarks!
>
> ---
>
> ### **Presentation**
>
> We sincerely apologize for the issues, such as heavy use of acronyms, inconsistent plot fonts, and the missing appendix, caused by time constraints. We will standardize all abbreviations, unify plot styles, and complete the Appendix in the final revision.
>
> ---
>
> ### **"All WC" refers to the dataset before the pre-filtering step**
>
> ---
> We thank the reviewer again for these constructive comments, which allowed us to explore the underlying principles of our method and significantly improve the theoretical grounding of our work!

---

> ### Author Response · Authors · 2025-11-25
> **Kindly Reminder**
>
> Dear Reviewer:
>
> Has our reply resolved your concern? If you have any further questions, please don't hesitate to ask, and we'll be happy to respond. We deeply appreciate your feedback, as it is invaluable for improving the quality of our work.

---

> > ### Author Response · Authors · 2025-11-27
> > **Kindly Reminder**
> >
> > Dear Reviewer:
> >
> > Has our reply resolved your concern? If you have any further questions, please don't hesitate to ask, and we'll be happy to respond. We deeply appreciate your feedback, as it is invaluable for improving the quality of our work.

---

### Official Review · Reviewer_tm2i · 2025-11-01

**Soundness:** 2
**Presentation:** 3
**Contribution:** 2
**Rating:** 4
**Confidence:** 4

**Summary:**

This paper addresses an important problem in LLM evaluation: how to construct an efficient, reliable, and unbiased benchmark for assessing the quality of automatic test case generation methods. The core contribution is a novel framework based on binary-matrix theory. This framework uses the matrix rank to simultaneously determine the minimal number of wrong codes required for evaluation and a theoretical upper bound on the number of test cases needed for full error pattern coverage. Based on this framework, the authors devised WrongSelect to find an approximate solution to this NP-hard problem, culminating in the construction of TC-Bench.

**Strengths:**

1. The paper tackles the crucial challenge of evaluating test case generation methods for code.
2. The author processes an efficient approximation algorithm combining pre-filtering and random-restart local search, WrongSelect. This provides a reasonable and effective solution to the NP-hard problem of selecting a maximally diverse diagnostic basis from a vast collection of wrong codes.
3. The authors release a compact and diverse benchmark, TC-Bench. By design, TC-Bench can reduce the computational cost of evaluation and is resistant to score inflation.

**Weaknesses:**

1. Insufficient discussion of related work. The paper could be strengthened by a more thorough discussion of related work that also leverages code-test properties. For instance, CodeT [1] or other approaches that also use binary matrices, such as B4 [2].

2. Limited theoretical justification for the rank-coverage duality. A central claim of the paper is that the matrix rank provides an upper bound on the minimum number of test cases required for fault coverage. While the paper lacks a formal proof or a detailed discussion to substantiate this claim. A deeper analysis would strengthen the claim beyond intuition.

3. Limitations of the chosen diversity metric. The paper employs Jaccard similarity to measure the overlap. It implicitly assumes that all GTs are of equal importance. In practice, some GTs may represent extremely rare and critical edge cases, while others are more trivial. The Jaccard metric cannot capture this weighted difference. Have the authors experimented with or considered other similarity metrics that could account for the varying significance of different test cases?

[1] Chen, Bei, et al. "CodeT: Code Generation with Generated Tests." The Eleventh International Conference on Learning Representations.

[2] Chen, Mouxiang, et al. "B4: Towards optimal assessment of plausible code solutions with plausible tests." Proceedings of the 39th IEEE/ACM International Conference on Automated Software Engineering. 2024.

**Questions:**

See Weaknesses. I will be glad to raise my score if the authors could provide a sufficient rebuttal.

---

> ### Author Response · Authors · 2025-11-19
> **Part1**
>
> We sincerely thank the reviewer for recognizing the value of our work in tackling the crucial challenge of evaluating test case generation!
>
> ---
> ### **Additional discussion of related work**
>
> We thank the reviewer for pointing out these works that share the idea of using the execution results of code on all test cases (the Code-Test Matrix) as signatures! CodeT assumes correct code behaviors are consistent, while incorrect code results are more diverse. It utilizes the signatures for clustering, selecting the consensus set that maximizes the product of cluster size and passed tests. B4 represents a significant advancement in this direction. It ingeniously frames the problem within a Bayesian framework, modeling the optimal selection strategy as a Maximum A Posteriori (MAP) estimation problem. Specifically, it calculates the posterior probability of the observed Code-Test Matrix given a hypothesized correct cluster and selects the solution with the highest probability. This principled and rigorous probabilistic modeling provides a highly effective method for identifying correct solutions.
>
> However, the properties and roles of our code and tests are different. TC-Bench operates on Ground Truth: our columns are Golden Test Cases (guaranteed correct), and our rows are Wrong Codes (guaranteed incorrect). **We view the matrix as a complete Error Space to calculate the Rank and Basis, thereby representing this error space in the most efficient way**. In contrast, the correctness of both code and tests is unknown in CodeT and B4. **For their objective of solution selection, utilizing the matrix as a collection of signatures is sufficient and effective, without necessitating linear algebra operations**. CodeT effectively performs signature matching, and B4 performs elegant and robust probabilistic modeling based on pass/fail counts. In their probabilistic context, the algebraic rank and basis are not the primary interpretative tools. For our problem, however, it is essential to utilize the matrix's algebraic properties to model the error space and apply Linear Algebra methods to solve our problem.
>
> In summary, we all effectively modeled and solved our respective problems through signatures. This demonstrates the immense value and research potential of the Code-Test Matrix! We have updated the relevant content in Appendix A.1 Related Work.
>
> ---
> ### **Why Rank is the Upper Bound of Test Cases**
>
> We regret that the theoretical intuition underlying the rank-coverage duality was not fully conveyed in the initial submission. The matrix rank plays a pivotal role because the row rank (number of independent error patterns) equals the column rank (number of independent diagnostic dimensions). **This means that in an error space defined by rank $R$, there are only $R$ linearly independent diagnostic dimensions. Any other test case is merely a linear combination of these basis dimensions and does not provide new information for distinguishing the existing error patterns.** Therefore, $R$ test cases are sufficient to distinguish all error patterns, making it a rigorous upper bound.
>
> Let us illustrate this with the concrete example below. The matrix has a rank of 3.
>
> ||t1|t2|t3|t4|
> |-|--|--|--|--|
> |wc1|1|0|1|0|
> |wc2|0|1|1|0|
> |wc3|0|1|1|0|
> |wc4|0|0|0|1|
>
> In this matrix, $t_1$ and $t_2$ are linearly independent, but $t_3$ is a linear combination of them (any wrong code failing $t_1$ or $t_2$ also fails $t_3$). Therefore, $t_3$ does not offer a new dimension for distinguishing errors. To fully diagnose the wrong codes in this matrix, the basis set $\{t_1, t_2, t_4\}$ is sufficient. The theoretical upper bound for the necessary test cases is thus 3.
>
> This framework addresses a critical flaw in previous evaluations where the number of test cases was arbitrary. Consequently, problems with small diagnostic dimensions were often "over-tested," inflating scores and masking differences between methods, while problems with large diagnostic dimensions were "under-tested," failing to reflect true capability. Using Rank as the budget ensures fairness: it allows simple problems to reveal performance gaps while ensuring complex problems are tested with sufficient depth.

---

> ### Author Response · Authors · 2025-11-19
> **Part2**
>
> Your insightful question also inspires a discussion on the lower bound. Theoretically, if we select $\{t_3, t_4\}$, we might distinguish all errors with just 2 test cases (since $t_3$ covers the failures of both $t_1$ and $t_2$). However, calculating this lower bound is complex because it relies on the existence of "composite" test cases like $t_3$ in the golden set. In practice, problem authors typically design test cases to target specific boundary conditions individually (like $t_1$ and $t_2$) rather than mixing them. Our experimental observations align with this: test cases that are "eliminated" (redundant) are typically combinations of a few simple basis cases, rather than complex combinations of many. Therefore, calculating the lower bound would be a more difficult and deeper problem.
>
> ---
>
> ### **Diversity Metric Can Show Test Case Importance in Evaluation**
>
> We respectfully submit that the basis selected via Jaccard similarity naturally reflects the varying value of test cases during the evaluation phase, even if error patterns are treated equivalently during the selection phase.  **Equivalence in selection does not imply equivalence in evaluation.** The matrix structure itself preserves the "importance" of test cases through the density of failure patterns (the number of '1's).
>
> For rare and critical test cases, a significant number of wrong codes will fail, resulting in a column with many '1's in the raw matrix. Regardless of how Jaccard similarity optimizes for diversity, the selected basis cannot avoid retaining a column with a high density of '1's for these critical patterns. Consequently, during evaluation, if a method generates such a rare test case, it will exclude a large number of wrong codes (match the many '1's in the basis), resulting in a high HackRate score.
>
> Conversely, for simple test cases, the raw matrix contains mostly '0's and a few '1's. Since Jaccard similarity encourages diversity (minimizing overlap), it tends to select rows that are distinct, maintaining the sparsity of '1's for these simple columns in the final basis. During evaluation, even if a method generates these simple test cases, they will exclude fewer wrong codes, resulting in a relatively lower score. Thus, without explicit manual weighting, **the basis constructed via Jaccard similarity automatically differentiates the value of test cases in the final evaluation scores**. Therefore, calculating the lower bound would be a more difficult and deeper problem.
>
> ---
>
> We thank the reviewer again for these valuable and constructive comments, which have helped us significantly improve our paper!

---

> ### Author Response · Authors · 2025-11-25
> **Kindly Reminder**
>
> Dear Reviewer:
>
> Has our reply resolved your concern? If you have any further questions, please don't hesitate to ask, and we'll be happy to respond. We deeply appreciate your feedback, as it is invaluable for improving the quality of our work.

---

> > ### Author Response · Authors · 2025-11-27
> > **Kindly Reminder**
> >
> > Dear Reviewer:
> >
> > Has our reply resolved your concern? If you have any further questions, please don't hesitate to ask, and we'll be happy to respond. We deeply appreciate your feedback, as it is invaluable for improving the quality of our work.

---

### Author Response · Authors · 2025-11-22

We express our deepest gratitude to all reviewers (Reviewers jxDY, tm2i, tPeL, D88H) for their exceptionally constructive, detailed, and insightful feedback! We are truly inspired by the depth of your reviews, which have been instrumental in identifying critical areas for improvement.

We have addressed all specific concerns in our individual responses to each reviewer. Furthermore, we have revised the paper to incorporate your suggestions. We believe that thanks to your guidance, the quality, rigor, and clarity of this work have been fundamentally elevated.

Below is a summary of the major revisions:
1. **Rigorous Comparison with Previous Work (Thanks to Reviewer jxDY)**: We have added an "apple-to-apple" comparison against previous benchmarks (e.g., TCG, TestCase-Eval) in Section 4 Discussion, demonstrating that TC-Bench effectively avoids both score inflation and under-representation.
2. **Clarification on Rank Upper Bound (Thanks to Reviewer tm2i)**: We have explicitly clarified the rationale in Section 4 Discussion explaining why the matrix rank serves as a compact upper bound for the minimum number of necessary test cases.
3. **Expanded Related Work (Thanks to Reviewer tm2i)**: We have updated Appendix A.1 Related Work to include a detailed comparison with other methodologies utilizing code-test matrices, highlighting the unique positioning of our framework.
4. **Comprehensive Case Study (Thanks to Reviewers tPeL and D88H)**: We have added a detailed case study in Appendix C Case Study, covering problem modeling, the WrongSelect algorithm, and **diagnostic insights on real-world scenarios**, to demonstrate the practical effectiveness of our method.
5. **Presentation Improvements (Thanks to Reviewer jxDY)**: We have refined the figures and typography throughout the paper to enhance readability and visual clarity.

We are eager to engage in further discussion and are committed to answering any additional questions promptly! Thank you once again for your time, effort, and invaluable contribution to improving our paper!

---

### Note · Authors · 2026-01-26

I have read and agree with the venue's withdrawal policy on behalf of myself and my co-authors.

---

> ### Note · Program_Chairs · 2026-01-26
>
> We approve the reversion of withdrawn submission.

---

### Meta-Review · Area_Chair_UFq5 · 2026-01-08

**Summary:**

The paper proposed an simple, effective and elegant method to address the score inflation issues for code evaluation. After reading the review comments and the rebuttal carefully, AC believes most of the concerns from the reviewers have been well addressed. The newly introduced TC-bench made significant contributions to the evaluation of code generation. The AC is glad to recommend acceptance. More details are below.

**Reviewer Concerns:**

- Reviewer tm2i: authors added more related work discussions, gave clear explanation of rank-coverage duality, and discussed the rationale of using Jaccard in a clever way. The response is sufficient and sound.
- Reviewer jxDY: the reviewer requested more apple-to-apple comparison, and the authors compared TC-bench with multiple previous benchmark using the same model. The additional table is convincing. The authors also verify the robustness of the algorithm.
- Reviewer tPeL: the concrete case studies are provided in a detailed way.
- Reviewer D88H: the review comment is short, but the authors responded the practical values and effectiveness in industry with the same case study.

**Reviewer Scores:**

All reviewers might increase or keep the scores unchanged. The overall scores will be positive.

---

### Decision · Program_Chairs · 2026-01-26

Accept (Poster)